# Computational models of O-LM cells are recruited by low or high theta frequency inputs depending on h-channel distributions

Vladislav Sekulić[1,2]*, Frances K Skinner[1,3]*

[1]Krembil Research Institute, University Health Network, Toronto, Ontario, Canada; [2]Department of Physiology, University of Toronto, Toronto, Ontario, Canada; [3]Departments of Medicine (Neurology) and Physiology, University of Toronto, Toronto, Ontario, Canada

**Abstract** Although biophysical details of inhibitory neurons are becoming known, it is challenging to map these details onto function. Oriens-lacunosum/moleculare (O-LM) cells are inhibitory cells in the hippocampus that gate information flow, firing while phase-locked to theta rhythms. We build on our existing computational model database of O-LM cells to link model with function. We place our models in high-conductance states and modulate inhibitory inputs at a wide range of frequencies. We find preferred spiking recruitment of models at high (4–9 Hz) or low (2–5 Hz) theta depending on, respectively, the presence or absence of h-channels on their dendrites. This also depends on slow delayed-rectifier potassium channels, and preferred theta ranges shift when h-channels are potentiated by cyclic AMP. Our results suggest that O-LM cells can be differentially recruited by frequency-modulated inputs depending on specific channel types and distributions. This work exposes a strategy for understanding how biophysical characteristics contribute to function.

*For correspondence: vlad. sekulic@gmail.com (VS); frances. skinner@gmail.com (FKS)

## Introduction

Despite many exciting developments in experimental techniques giving rise to more detailed and extensive datasets, it is far from clear how to harness these advances to increase our understanding of brain functioning. This is largely due to the multiple spatial and temporal scales that are known to exist in the brain (*Buzsáki, 2006*), making it challenging to understand how biophysical, cellular, circuit and behavioural aspects are linked. What is clear is that theoretical modeling must play a role to achieve an understanding (*Churchland and Abbott, 2016*). However, using models to determine, for instance, what biophysical details matter for circuit performance, can become obscured (*Gjorgjieva et al., 2016*). This is largely because the rationale for choosing what level of detail to include, and how model parameters are related to variable biological data, is not always provided. In this work, we take advantage of our previous cellular models, where the rationale and parameter choices are provided in detail, to suggest how underlying biophysical conductances in a particular cell type could specifically contribute to behaviourally relevant theta rhythms in the hippocampus.

During active exploration and REM sleep, theta rhythms are a prominent activity in the hippocampus, extending from about 4 to 12 Hz (*Colgin, 2013*). Two types of theta activity have been previously described, corresponding to high (7–12 Hz) or low (4–7 Hz) frequencies that are, respectively, atropine-resistant (Type 1) or atropine-sensitive (Type 2) (*Buzsáki, 2002*; *Kramis et al., 1975*). Moreover, recent work indicates that social or fearful stimuli elicit high or low theta rhythms respectively

**eLife digest** Neurons transmit information using electrical activity. Whereas electrical currents in wires consist of moving electrons, those in neurons are made up of charged particles called ions. These flow into and out of the cells through specialized channels in the outer membrane. Changes in ion channels can disrupt brain activity. However, unravelling the interactions between molecules that give rise to particular behaviours or diseases is challenging in living animals. Computer models can simplify this task, but only if the models are accurate copies of living systems. It is never possible to obtain a completely accurate model, so instead the goal is to make sure that any understanding derived from the models can guide new experiments, and that models are refined appropriately based on the results of these new experiments.

Sekulić and Skinner used computer modelling to explore how ion channels affect the properties of a type of neuron called the oriens lacunosum/moleculare (O-LM) cell. These are found in a region of the brain called the hippocampus, which is involved in learning and memory. When animals explore their environment, the O-LM cells fire in synchrony with other cells in the hippocampus. Hippocampal cell populations typically fire either between 4 and 7 times per second, known as low theta, or 7 to 12 times per second, called high theta. Low theta firing supports the processing of emotions, whereas high theta helps animals form a mental map of their surroundings.

Sekulić and Skinner wanted to know how the distribution of ion channels in O-LM cells – particularly a subtype called h-channels – affects whether the cells take part in theta firing. The task was made possible by the availability of a database of computer models of O-LM cells, each featuring a different distribution of ion channels. Sekulić and Skinner exposed each model to patterns of activity simulating those in the brain of an animal exploring its environment. The results revealed that specific combinations and distributions of ion channels predispose, or "tune", O-LM cells to participate in either low or high theta, and thus emotional or spatial learning in the hippocampus.

The next step is to test the predictions of the models experimentally. Studies should examine whether the distribution of ion channels in O-LM cells does indeed predispose them to fire at particular frequencies, as the models suggest. A further question is how targeting O-LM cells in freely moving animals would affect spatial and emotional memory.

(*Tendler and Wagner, 2015*). Furthermore, in vivo studies show that rhythmic activity of parvalbumin-positive (PV[+]) neurons in the medial septum-diagonal band of Broca (MS-DBB) can entrain the hippocampus at theta (*Hangya et al., 2009*), and that PV[+] MS-DBB neurons are known to specifically target interneurons in the hippocampus (*Gulyás et al., 1990*; *Garrett et al., 2014*). Given all of this, the mechanisms underlying the generation of theta rhythms in the hippocampus are likely multi-pronged.

Inhibitory interneurons in the hippocampus and cortex are diverse in many ways that include their morphology, synaptic targets, and molecular markers (*Freund and Buzsáki, 1996*; *Tremblay et al., 2016*). These specialized interneuron properties are likely to be functionally important since different interneurons are activated in distinct ways during network rhythms and behavioural tasks (*Kepecs and Fishell, 2014*; *Klausberger and Somogyi, 2008*). Oriens-lacunosum/moleculare (O-LM) cells, a particular interneuron type, fire during the trough of theta rhythms as recorded during local field potentials in the pyramidale layer of the CA1 region of hippocampus (*Klausberger et al., 2003*; *Varga et al., 2012*). O-LM cells target the distal dendrites of pyramidal cells and are part of a feedback inhibitory loop that gates information flow in CA1 (*Leão et al., 2012*; *Maccaferri, 2005*). Moreover, they are known to express hyperpolarization-activated inward channels (h-channels) (*Maccaferri and McBain, 1996*) that would allow them to exhibit post-inhibitory rebound spiking, a cellular component of circuit dynamics (*Getting, 1989*). Given the specific targeting of O-LM cells by inhibitory PV[+] MS-DBB neurons (*Garrett et al., 2014*), it is possible that such inputs at theta frequencies could allow O-LM cells to contribute to in vivo theta rhythms via post-inhibitory rebound spiking mechanisms.

The contribution of h-channels to subthreshold resonance and spiking output measures has been studied extensively in hippocampal pyramidal neurons (*Hu et al., 2002*, *2009*; *Narayanan and Johnston, 2008*; *Vaidya and Johnston, 2013*). In comparison, few studies have examined the contribution of h-channels in hippocampal interneuron function. *Pike et al. (2000)* found that putative O-LM cells exhibit a peak subthreshold membrane oscillation at 5 Hz, within the theta range. *Maccaferri and McBain (1996)* showed that upon blockade of h-channels, spontaneous 8 Hz firing in O-LM cells in vitro was substantially reduced. Accordingly, the presence of h-channels in O-LM cells was incorporated as a critical feature of a proposed mechanism of how theta rhythms could be generated in CA1 circuits that includes O-LM cells and fast-spiking interneurons (*Rotstein et al., 2005*). This mechanism considers O-LM cells as theta pacemakers and subsequent use of this model mechanism (*Gloveli et al., 2005*; *Wulff et al., 2009*) incorporated reduced single-compartment versions of our original O-LM multi-compartment model with h-channels in the soma only (*Saraga et al., 2003*).

However, further work has shown that in a more depolarized state, h-channels in O-LM cells are not expected to contribute to subthreshold oscillations at theta (*Zemankovics et al., 2010*). Furthermore, when placed in a high-conductance (in vivo-like) state using dynamic clamp, O-LM cells do not function as theta spiking pacemakers despite exhibiting subthreshold resonance at theta when h-current was enhanced using dynamic clamp (*Kispersky et al., 2012*). On the other hand, O-LM cells in that study did respond preferentially to 8 Hz theta frequency-timed inputs. This response was maintained even when h-channels were blocked, and was found to be dependent on afterhyperpolarization dynamics, presumably from outward potassium currents. However, these results were determined using dynamic clamp to inject artificial synaptic currents into the soma only, thereby ignoring the potential contribution of dendritic conductances, including h-channels, on the integration of synaptic inputs and formation of theta-modulated outputs.

The integration of synaptic input in neurons depends on the complement and distribution of dendritic voltage-gated channels as well as the pattern of synaptic inputs onto dendrites (*Stuart et al., 2008*; *Narayanan and Johnston, 2012*). For instance, non-uniform dendritic densities of h-channels in excitatory pyramidal cells allow for the integration of spatially disparate excitatory inputs on the dendritic tree to nevertheless arrive synchronously in the soma (*Magee and Cook, 2000*; *Williams and Stuart, 2000*). This highlights the importance of investigating whether dendritic inputs onto O-LM cells, modulated at theta frequencies, might better recruit h-channels to generate spiking activity at theta frequencies. Although the presence of h-channels in O-LM cells is clear, their distribution is unknown, and our original (*Saraga et al., 2003*) and later (*Lawrence et al., 2006b2006b*) multi-compartment O-LM cell models focused on somatic h-channel distributions only given the available experimental data. Further, it is now known that a given cell type can have quite different ion channel conductance densities for the same channel type and robustly maintain cell type-specific output (*Marder and Goaillard, 2006*). From a modeling perspective, the consideration of multiple models for a given cell type can capture and help with understanding ion channel conductance variability from functional perspectives (*Marder and Taylor, 2011*). Considering this, in previous work we built populations of O-LM cell models which had either h-channels in their soma only or in their soma and dendrites (*Sekulić et al., 2014*). With these models, we uncovered co-regulations between different channel conductances that included h-channels and two outward potassium channels.

In this paper we take advantage of our previously developed multi-compartment O-LM cell models to examine the synaptic and intrinsic conditions under which O-LM cells may be recruited to fire at theta frequencies in high-conductance states. In particular, we are interested in assessing the contribution of dendritic synaptic inputs as well as whether dendritic distributions and balances of conductances are needed to optimally recruit theta frequency firing. We find that our O-LM model cells are preferentially recruited at theta frequencies that can be at low or high theta ranges. Furthermore, this differential recruitment depends on h-channel distribution and its balance with the presence of the slow delayed-rectifier potassium channels in dendrites. Finally, recruitment at high theta is enhanced by shifting the h-channel voltage dependency of activation, as for instance via elevation of intracellular cyclic AMP (*Biel et al., 2009*) which, in O-LM cells, can be mediated by noradrenergic modulation (*Maccaferri and McBain, 1996*).

In summary, our work leverages a database of multi-compartment models to thoroughly examine regimes of synaptic and intrinsic voltage-gated conductances required to allow O-LM cells to be recruited at rhythmic activities in high-conductance states. Our work has implications and predictions

for experimental investigations into O-LM cell activity during theta rhythms in vivo. Also, our work provides a strategy for examining cell-specific contributions to behaviour.

## Results

Using multi-compartment models of oriens-lacunosum/moleculare (O-LM) cells, we examined spiking differences using somatic or somatodendritic synaptic inputs, with a focus on the more biologically realistic somatodendritic synaptic input context within our models. We explored the responses of our O-LM cell models given modulated inhibitory synaptic input at different frequencies and in consideration of whether h-channels were present in the soma only or in the soma and dendrites. We refer to h-channels as 'H' throughout (see Materials and methods).

### Obtaining a population of multi-compartment models representing O-LM cells with different H distributions

We selected 32 models from our previously developed database of O-LM cell models, with 16 models expressing somatic H and 16 models expressing somatodendritic H distributions (*Sekulić et al., 2014*). We note that while it is clear that h-channels are present in O-LM cells (*Maccaferri and McBain, 1996*; *Zemankovics et al., 2010*), how they are distributed is unknown at present. We had previously investigated the effect of introducing non-uniform distributions of dendritic H and found that, as long as total H conductance across the membrane was conserved, models with either uniform or non-uniform dendritic distributions could both account for experimental recordings from O-LM cells (*Sekulić et al., 2015*). We thus used uniform distributions of dendritic H in this work as present in our original model database (*Sekulić et al., 2014*).

We extracted highly-ranked models from our database such that all models of a particular distribution (somatic or somatodendritic H) had the same maximum conductance density; however, maximum conductance densities of the other channels were different across models (*Table 1*). This ensured that differences in the underlying H parameter would not be a confounding factor when subsequently examining how synaptic inputs with varying modulation frequencies affect model outputs. As done previously, we fitted the H activation time constant as well as the passive membrane properties such that the extracted models would specifically capture the sag response of the experimental data (*Sekulić et al., 2015*). See Materials and methods, *Table 2* and *Figure 1—figure supplement 1* for further details. We considered these 32 models as representative O-LM cells for the purposes of the present work.

### Somatodendritic inhibitory synaptic inputs more effectively entrain O-LM cell spiking than somatic inputs

To understand the responses of O-LM cells in vivo, we situated our models in high-conductance states (*Destexhe, 2007*). It has been shown that in vivo states can be produced using single or multi-compartment models as well as being re-created in vitro by use of the dynamic clamp (*Destexhe et al., 2001*, *2003*). In previous work, in vivo recordings from neocortical cells were available to directly estimate synaptic parameters (i.e., release sites, numbers and rates, correlations, etc.) to capture this barrage of synaptic activities in models (*Destexhe and Paré, 1999*). This is not the case for O-LM cells (or any other cell type) and we did not consider it appropriate to introduce these details at this time without having further experimental constraints. Instead, we used similar synaptic parameters as those chosen by *Kispersky et al. (2012)*, who created high-conductance states in O-LM cells in vitro using dynamic clamp.

The in vivo-like, or high-conductance states in our models were generated in the following way. Excitatory and inhibitory artificial synaptic inputs were distributed either in the soma of all O-LM cell models, or spread across the somatodendritic tree (*Figure 1A–C*) using uncorrelated inputs and the same rates as used by *Kispersky et al. (2012)*. The distributions of subthreshold fluctuations of somatic $V_m$ depended on model parameters and distribution of synaptic inputs. Thus, peak synaptic conductances were scaled on a per-model basis, resulting in two sets of peak conductances, one for each distribution of synaptic inputs, to ensure that prior to input modulation, all models exhibited similar baseline firing characteristics of ~2.5 Hz output, with fluctuations of approximately 2 mV (*Figure 1D–F*, *Figure 1—figure supplement 2*). See Materials and methods for further details.

**Table 1.** Parameter values for models used in this work. Parameters taken from the database include the ion channel maximum conductance densities and the morphology (cell 1 or cell 2; see **Figure 1B**), H channel distribution ($H_S$ – somatic H; $H_{SD}$ – somatodendritic H, uniformly distributed). Units for maximum conductance densities are in $pS/\mu m^2$.

**Somatic H models**

| Rank | 326 | 556 | 613 | 620 | 689 | 723 | 755 | 769 | 26 | 31 | 39 | 43 | 45 | 60 | 67 | 68 |
|------|-----|-----|-----|-----|-----|-----|-----|-----|----|----|----|----|----|----|----|----|
| cell | 1 | 1 | 1 | 1 | 1 | 1 | 1 | 1 | 2 | 2 | 2 | 2 | 2 | 2 | 2 | 2 |
| hD | 0 | 0 | 0 | 0 | 0 | 0 | 0 | 0 | 0 | 0 | 0 | 0 | 0 | 0 | 0 | 0 |
| H | 0.5 | 0.5 | 0.5 | 0.5 | 0.5 | 0.5 | 0.5 | 0.5 | 0.5 | 0.5 | 0.5 | 0.5 | 0.5 | 0.5 | 0.5 | 0.5 |
| Nad | 117 | 117 | 117 | 117 | 117 | 117 | 117 | 117 | 117 | 117 | 117 | 117 | 117 | 117 | 117 | 117 |
| Nas | 107 | 220 | 107 | 220 | 107 | 60 | 60 | 107 | 107 | 107 | 107 | 60 | 107 | 60 | 107 | 107 |
| Kdrf | 215 | 215 | 215 | 215 | 215 | 215 | 215 | 215 | 215 | 215 | 215 | 215 | 215 | 215 | 215 | 215 |
| Kdrs | 2.3 | 2.3 | 2.3 | 2.3 | 2.3 | 2.3 | 2.3 | 2.3 | 2.3 | 2.3 | 2.3 | 2.3 | 2.3 | 2.3 | 2.3 | 2.3 |
| KA | 2.5 | 32 | 32 | 32 | 2.5 | 32 | 2.5 | 2.5 | 2.5 | 32 | 2.5 | 2.5 | 32 | 32 | 32 | 2.5 |
| CaT | 2.5 | 5 | 2.5 | 5 | 2.5 | 5 | 5 | 1.25 | 1.25 | 5 | 2.5 | 2.5 | 5 | 2.5 | 2.5 | 5 |
| CaL | 50 | 50 | 25 | 25 | 25 | 25 | 25 | 25 | 25 | 25 | 25 | 25 | 50 | 12.5 | 50 | 50 |
| AHP | 11 | 5.5 | 2.75 | 5.5 | 5.5 | 5.5 | 11 | 5.5 | 11 | 5.5 | 11 | 2.75 | 2.75 | 2.75 | 2.75 | 11 |
| M | 0.375 | 0.375 | 0.375 | 0.375 | 0.75 | 0.375 | 0.375 | 0.75 | 0.375 | 0.375 | 0.375 | 0.375 | 0.375 | 0.375 | 0.375 | 0.375 |

**Somatodendritic H models**

| Rank | 225 | 356 | 913 | 1230 | 1520 | 2050 | 2173 | 2286 | 6 | 34 | 37 | 49 | 57 | 92 | 96 | 109 |
|------|-----|-----|-----|------|------|------|------|------|---|----|----|----|----|----|----|-----|
| cell | 1 | 1 | 1 | 1 | 1 | 1 | 1 | 1 | 2 | 2 | 2 | 2 | 2 | 2 | 2 | 2 |
| hD | 1 | 1 | 1 | 1 | 1 | 1 | 1 | 1 | 1 | 1 | 1 | 1 | 1 | 1 | 1 | 1 |
| H | 0.1 | 0.1 | 0.1 | 0.1 | 0.1 | 0.1 | 0.1 | 0.1 | 0.1 | 0.1 | 0.1 | 0.1 | 0.1 | 0.1 | 0.1 | 0.1 |
| Nad | 230 | 230 | 230 | 230 | 230 | 230 | 230 | 230 | 230 | 230 | 230 | 230 | 230 | 230 | 230 | 230 |
| Nas | 107 | 220 | 107 | 60 | 60 | 60 | 107 | 220 | 60 | 107 | 107 | 107 | 107 | 60 | 60 | 107 |
| Kdrf | 506 | 506 | 506 | 506 | 506 | 506 | 506 | 506 | 506 | 506 | 506 | 506 | 506 | 506 | 506 | 506 |
| Kdrs | 42 | 42 | 42 | 42 | 42 | 42 | 42 | 42 | 42 | 42 | 42 | 42 | 42 | 42 | 42 | 42 |
| KA | 2.5 | 2.5 | 2.5 | 2.5 | 2.5 | 2.5 | 2.5 | 2.5 | 2.5 | 2.5 | 2.5 | 2.5 | 2.5 | 2.5 | 32 | 32 |
| CaT | 1.25 | 2.5 | 1.25 | 1.25 | 2.5 | 1.25 | 2.5 | 1.25 | 5 | 5 | 1.25 | 5 | 1.25 | 2.5 | 1.25 | 2.5 |
| CaL | 25 | 50 | 50 | 25 | 12.5 | 25 | 25 | 25 | 25 | 25 | 12.5 | 12.5 | 25 | 50 | 25 | 50 |
| AHP | 5.5 | 2.75 | 5.5 | 2.75 | 5.5 | 5.5 | 5.5 | 5.5 | 2.75 | 2.75 | 2.75 | 2.75 | 11 | 5.5 | 5.5 | 5.5 |
| M | 0.75 | 0.375 | 0.75 | 0.375 | 0.375 | 0.375 | 0.375 | 0.75 | 0.375 | 0.375 | 0.75 | 0.375 | 0.375 | 0.375 | 0.375 | 0.375 |

The subthreshold $V_m$ activity of models with no input modulation showed significantly less variability in $V_m$ fluctuations for somatodendritic inputs compared to that of somatic inputs (**Figure 1E**). Importantly, despite the differences in subthreshold fluctuations between somatic and somatodendritic inputs, the baseline firing prior to modulation was held at approximately 2.5 Hz for models in both input conditions (**Figure 1F**). This difference between somatic and somatodendritic synaptic input locations can be understood by considering the temporal patterning of individual excitatory and inhibitory synaptic events as seen at the soma. With independently generated (i.e., uncorrelated) excitatory and inhibitory synaptic inputs spread across the dendritic tree, there are fewer trains of consecutive excitatory or inhibitory inputs that summate to produce larger amplitudes at the soma, compared to when only somatic input trains are present. Therefore, synaptic inputs that are spread across the dendritic tree tend to produce smaller $V_m$ fluctuations compared to when only one somatic input location is present. We also found a significantly more depolarized mean $V_m$ for models with somatodendritic H in the case of somatodendritic inputs, compared to somatic H models, which exhibited no statistically significant change in mean $V_m$ across synaptic input locations (**Figure 1D**). Overall, this depolarized $V_m$ along with the reduced fluctuations (**Figure 1E**) with

**Table 2.** Additional parameters re-fitted as per *Sekulić et al. (2015)* to improve h-channel activation kinetics and passive properties. Shown here are the specific membrane resistivity, $R_m$, the specific membrane capacitance, $C_m$., and bias current needed to keep model somatic $V_m$ at −74 mV as per the experimental data used for fitting. However, the bias current was not used in any of the high-conductance synaptic input simulations. For the fitted H channel steady-state activation function, see *Figure 1—figure supplement 1*.

| Somatic H models | | | | | Somatodendritic H models | | | |
|---|---|---|---|---|---|---|---|---|
| Rank | Cell | $R_m$ ($\Omega \cdot cm^2$) | $C_m$ ($\mu F/cm^2$) | $I_{bias}$ (pA) | Rank | Cell | $R_m$ ($\Omega \cdot cm^2$) | $C_m$ ($\mu F/cm^2$) | $I_{bias}$ (pA) |
| 326 | 1 | 80,932 | 0.5119 | −6.79 | 225 | 1 | 138,328 | 0.6603 | −10.3 |
| 556 | 1 | 90,251 | 0.4981 | −0.932 | 356 | 1 | 122,359 | 0.6515 | −11.3 |
| 613 | 1 | 89,118 | 0.5282 | −1.57 | 913 | 1 | 130,845 | 0.6524 | −10.3 |
| 620 | 1 | 90,099 | 0.5025 | −0.932 | 1230 | 1 | 131,079 | 0.6574 | −11.2 |
| 689 | 1 | 79,102 | 0.5069 | −8.19 | 1520 | 1 | 130,763 | 0.6520 | −10.5 |
| 723 | 1 | 90,289 | 0.4950 | −0.846 | 2050 | 1 | 131,588 | 0.6528 | −10.5 |
| 755 | 1 | 80,939 | 0.5058 | −6.83 | 2173 | 1 | 129,748 | 0.6505 | −10.5 |
| 769 | 1 | 79,183 | 0.5057 | −8.16 | 2286 | 1 | 130,703 | 0.6547 | −10.4 |
| 26 | 2 | 64,872 | 1.046 | −15.2 | 6 | 2 | 71,586 | 1.086 | −7.0 |
| 31 | 2 | 68,312 | 1.058 | −12.3 | 34 | 2 | 68,585 | 1.083 | −7.04 |
| 39 | 2 | 63,968 | 1.060 | −15.2 | 37 | 2 | 71,079 | 1.086 | −6.9 |
| 43 | 2 | 63,041 | 1.042 | −16.4 | 49 | 2 | 71,129 | 1.083 | −7.04 |
| 45 | 2 | 66,584 | 1.058 | −12.7 | 57 | 2 | 71,750 | 1.082 | −5.76 |
| 60 | 2 | 67,959 | 1.048 | −12.6 | 92 | 2 | 71,891 | 1.081 | −6.56 |
| 67 | 2 | 68,083 | 1.047 | −12.7 | 96 | 2 | 72,746 | 1.066 | −2.13 |
| 68 | 2 | 64,339 | 1.053 | −15.2 | 109 | 2 | 75,364 | 1.061 | −2.18 |

somatodendritic inputs is related to choosing synaptic parameters that allow the models to be brought into the ~2.5 Hz firing regime (*Figure 1F*). In other words, since the somatodendritic H models exhibited smaller $V_m$ fluctuations, their mean $V_m$ necessarily needed to be more depolarized in order for the fluctuations to cross the firing threshold frequently enough to produce, on average, approximately 2.5 Hz firing with no input modulation.

Modulation of the inhibitory synaptic input trains was performed at various frequencies (0.5–30 Hz, *Figure 1—figure supplement 3*), and the spiking output of models was assessed using the power ratio, i.e., the power spectral density (PSD) peak at the modulation frequency divided by the value at 0 Hz frequency. We found significantly higher power ratios for all tested frequencies and all models with somatodendritic inputs compared to somatic inputs, regardless of the dendritic H distribution in the models (*Figure 1G*). We note that the direct comparison between somatic and somatodendritic inputs can be done since the spiking of models with no input modulation was kept at around 2.5 Hz for both synaptic input cases (*Figure 1F*).

Because distributing synaptic inputs across the somatodendritic tree allowed O-LM cell models to be better entrained at oscillatory inputs across a wide range of frequencies relative to somatic only synaptic inputs, we focus on somatodendritic inputs for the remainder of this work. Further, somatodendritic inputs are more biologically relevant compared to somatic only inputs since synaptic inputs also target dendrites. We note that our goal was not to compare somatic and somatodendritic synaptic input scenarios per se, and as such we did not try to adjust excitatory or inhibitory rates for these different scenarios. Rather, we simply adjusted the synaptic weights to ensure similar baseline spiking frequencies (see Materials and methods).

## Preferred spiking resonance and firing precision frequency depends on H distribution and can be partitioned into low and high theta frequency responses

We assessed the spiking response of our models as a function of modulated inhibitory input frequency, using the power ratio as well as the rotation number, defined as the average number of

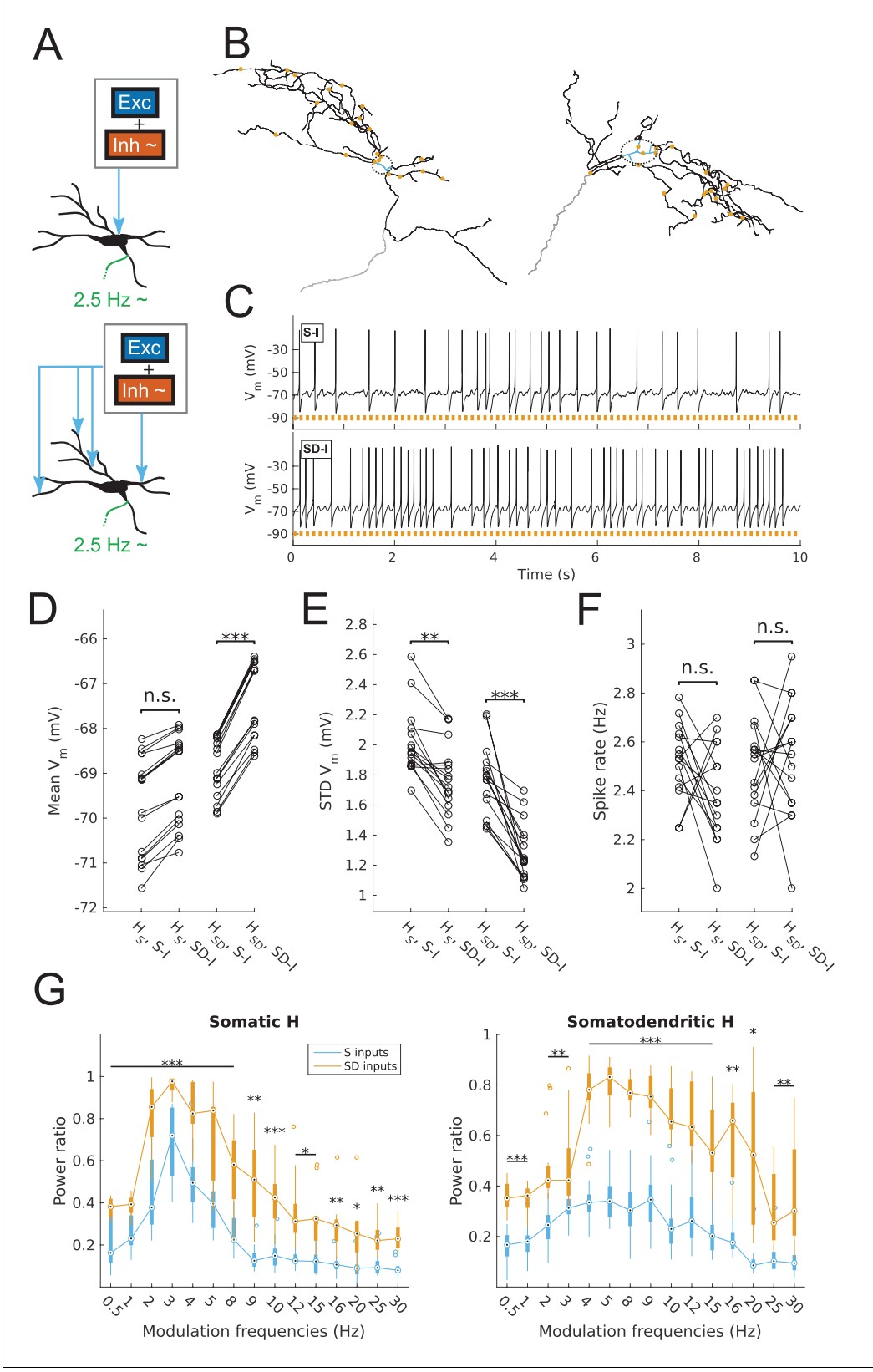

**Figure 1.** Spiking output of O-LM cell models with somatic or somatodendritic artificial synaptic inputs. (**A**) Schematic of the virtual protocol for somatic (top) and somatodendritic (bottom) synaptic inputs. Representative O-LM model morphology shown with soma and dendrites (black) and truncated axon (green). Excitatory and

*Figure 1 continued*

inhibitory Poisson process synaptic inputs shown in grey box; note that each synaptic input location has an independent excitatory and inhibitory Poisson process. Synaptic inputs are tuned to produce approximately 2.5 Hz output prior to input modulation. Only the inhibitory inputs are modulated ('~' symbol). See Materials and methods for full details. (B) Locations of synaptic inputs when distributed along soma and dendrites (SD inputs case; see Materials and methods) for both O-LM cell morphologies (cell 1, left; cell 2, right; dendrites in black; soma in blue with surrounding dashed black ellipse; truncated axon in grey; synaptic input locations in orange). The case with somatic inputs contains only one input location at the middle of the soma (not shown). (C) Example somatic voltage traces from a model with somatodendritic H ($H_{SD}$, rank 109) showing spike trains with 8 Hz modulation for somatic inputs (S-I, top) and somatodendritic inputs (SD-I, bottom). Orange bars at bottom denote peak phase of modulation at 8 Hz (see Materials and methods). (D) Mean subthreshold $V_m$ for models with somatic ($H_S$) vs. somatodendritic ($H_{SD}$) H distributions and somatic inputs (S-I) vs. somatodendritic (SD-I) inputs, all with no modulation ($H_S$: no significant difference, n = 16; $H_{SD}$: ***p<0.001, n = 16; Wilcoxon rank sum test performed for both cases). (E) Fluctuations of subthreshold $V_m$ of models with $H_S$ vs. $H_{SD}$ and S-I vs. SD-I, all with no modulation, as measured by the standard deviation of subthreshold $V_m$. ($H_S$: **p<0.01, n = 16, Wilcoxon rank sum test; $H_{SD}$: ***p<0.001, n = 16, paired *t*-test). (F) Spike rates of models with $H_S$ vs. $H_{SD}$ and S-I vs. SD-I, all with no modulation, with no significant difference for both $H_S$ and $H_{SD}$ cases (n.s., n = 16 each; paired *t*-test performed for both). (G) Power ratio, or ratio of power at modulation frequency to 0 Hz frequency, for models with $H_S$ (somatic H, left) and $H_{SD}$ (somatodendritic H, right), with S-I (blue) and SD-I (orange). Power ratios are significantly higher for all modulation frequencies for both $H_S$ and $H_{SD}$ models. Statistical tests used were two-way repeated measures ANOVA performed separately on the populations of $H_S$ and $H_{SD}$ models, between all modulation frequencies crossed with input location ($H_S$: $F_{(1,15)}$ = 13.55, p<0.001, n = 16; $H_{SD}$: $F_{(1,15)}$ = 5.027, p=0.017, n = 16; Huynd-Feldt correction reported for both tests). Boxplots denote median of power ratios at the circle; 25th and 75th percentiles are denoted by the extent of the thick coloured bars; full extent of data denoted by thin lines extending from the bars, with outliers shown as coloured open circles. Outliers are defined as points outside of $q_3 \pm 2.7\sigma(q_3 - q_1)$, where $q_1$ and $q_3$ are the 25th and 75th percentiles, respectively (see *boxplot* function in MATLAB). Stars denote level of significance from Tukey's post-hoc tests, with p<0.05 (*), p<0.01 (**), and p<0.001 (***). Multiple modulation frequencies sharing the same level of significance are connected with a horizontal bar above. When there are no stars at a particular modulation frequency, this denotes no statistical difference was found between the two populations. If no statistical difference was found across all frequencies, a horizontal bar across all x-axis values is placed with a label of 'n.s.' on top, meaning 'not significant'. All subsequent boxplots in later figures share this design.

The following figure supplements are available for figure 1:

**Figure supplement 1.** Model spiking responses before and after optimizing passive properties and h-channel kinetics.

**Figure supplement 2.** Synaptic parameters for models.

**Figure supplement 3.** Patterning of synaptic inputs.

---

spikes per input cycle (see Materials and methods). For the baseline case with no modulation, models exhibited averaged responses of 2.5 Hz firing as per our simulation design described above (*Figure 1F* and *Figure 2A*, top). Upon modulation of the inhibitory inputs, we found broad differences in response properties of models depending on H distribution. For instance, both somatic and somatodendritic H models could follow 3 Hz input reliably (*Figure 2A*, middle), whereas only somatodendritic H models could follow 8 Hz inputs (*Figure 2A*, bottom). A complete examination of responses showed that models with somatic H exhibited a peak in spiking resonance at 3 Hz with sharp drop-off at lower and higher frequencies (*Figure 2B*, left), whereas models with somatodendritic H showed a peak at 4 Hz with a broader response at theta frequencies (power ratio >0.6 at 4–10 Hz, *Figure 2C*, left). We used rotation number, or average number of spikes per input cycle, as a supplementary measure to power ratio. We found that rotation number provided a more direct and understandable measure of spike recruitment. We considered models with a rotation number between 0.5 and 1 as being recruited to fire at a majority of input cycles at the given modulated frequency (see Discussion). Examination of rotation number for both types of models demonstrated that somatic H models were recruited to fire at a majority of input cycles for 2–5 Hz input

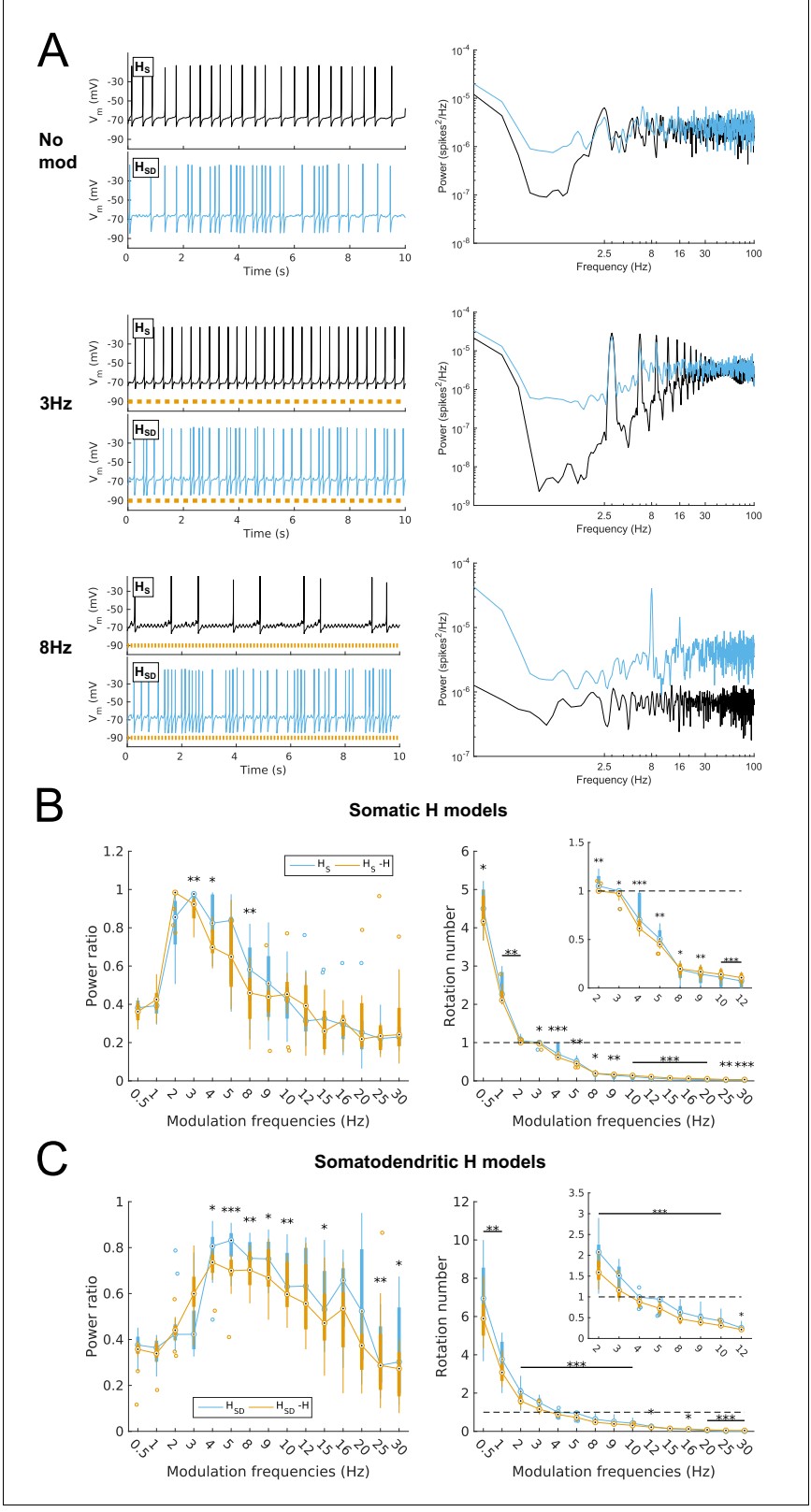

**Figure 2.** Effects of dendritic H distribution and H block on recruitment of O-LM model spiking in response to modulated somatodendritic inhibitory input. (**A**) Somatic $V_m$ traces for example somatic H model ($H_S$, black) and somatodendritic H model ($H_{SD}$, blue) under various modulation conditions (top – no modulation; middle – 3 Hz modulation; bottom – 8 Hz modulation). With modulated inputs, orange bars at bottom denote phase of peak

*Figure 2 continued on next page*

*Figure 2 continued*

modulation at the specified frequency (see Materials and methods). Power spectrum density (PSD) plots shown to the right of each output trace. (**B, C**) Power ratios (left) and rotation numbers (right) under different modulation frequencies for models with somatic H (**B**) and somatodendritic H (**C**) in control and H blocked ('-H') conditions. Insets in rotation number plots show zoomed portion in the theta range (2–12 Hz). Statistical test used was repeated measures ANOVA for the populations of $H_S$ and $H_{SD}$ models between all modulation frequencies crossed with H block condition (power ratios $H_S$: $F_{(1,15)}$ = 2.23, p=0.013, n = 16; $H_{SD}$: $F_{(1,15)}$ = 2.89, p=0.017, n = 16; rotation numbers $H_S$: $F_{(1,15)}$ = 27.94, p<0.001, n = 16; $H_{SD}$: $F_{(1,15)}$ = 10.35, p=0.006, n = 16; Huynd-Feldt correction reported for all tests). Boxplot annotations as per *Figure 1G* legend.

The following figure supplements are available for figure 2:

**Figure supplement 1.** Effect of blocking H on sub- and suprathreshold measures.

**Figure supplement 2.** Partitioning of spiking responses of O-LM models into high and low theta when using 'H leak' instead of H block.

(*Figure 2B*, right) whereas somatodendritic H models fired at a majority of input cycles for 4–9 Hz (*Figure 2C*, right).

We next tested whether the differential responses of models would be affected with H block. We found that upon blockade of H, there was a prominent hyperpolarization of $V_m$ that resulted in no firing of models under any input frequency, or even with no modulation, using our previously tuned synaptic inputs (*Figure 2—figure supplement 1A–C*). In other words, all models were shifted away from the fluctuation-driven firing regime into a quiescent, hyperpolarized state due to the lack of inward current from H. To adequately test whether models without H could still exhibit spiking resonance, we needed to bring the models back into the high-conductance state. One option was to add a depolarizing current from a virtual somatic current clamp. Due to space clamp issues, however, the effects of the clamp on distal portions of the dendritic tree would be less pronounced, thus potentially biasing the results for one category of models over another (either somatic or somatodendritic H). Thus, we decided to adjust the synaptic conductances directly, which is also a more physiologically plausible way to control for differences in membrane conductance due to blockade of voltage-gated channels. Because modulation was performed on the inhibitory inputs, however, changing the peak inhibitory conductances could introduce a separate confounding factor. Namely, any resulting change in spiking power could either be attributed to H block, or to the change in the effectiveness of the input modulation itself. Thus, to avoid this confound, we kept the peak inhibitory conductances fixed. Instead, we only increased the peak excitatory conductance on a per-model basis to bring the models with H block back into the fluctuation-driven regime of firing at approximately 2.5 Hz prior to modulation while simultaneously maximizing the standard deviation of subthreshold $V_m$ fluctuations (*Figure 2—figure supplement 1*; *Figure 1—figure supplement 2* ). All subsequent references to models with H block (or '-H' in legends) refer to the models with these adjusted synaptic conductances.

Upon input modulation, we found that somatic H models with H block showed a modest but statistically significant reduction in power ratios at 3–4 Hz and 8 Hz and as well as in rotation number across all frequencies (*Figure 2B*). Models with somatodendritic H with H block exhibited a modest but statistically significant reduction of power ratio at 4–10 Hz and higher frequencies, as well as in rotation number at 2–10 Hz (*Figure 2C*). Additional control simulations in which H was replaced with an artificial leak ('H leak') conductance to maintain baseline firing without manipulating synaptic conductances yielded similar results, particularly in rotation numbers (*Figure 2—figure supplement 2*). Thus, blocking H resulted in modest impairment of spiking resonance at a broad range of theta frequencies, with additional reductions in the high theta range for somatodendritic H models. However, the partitioning of preferred spiking responses of somatic and somatodendritic H models to low and high theta timed inputs, respectively, was preserved despite H block.

Finally, we studied the ability of model firing to be recruited at particular phases of the modulated input. This was done by calculating the vector strength (VS) or synchronization index, a measure of firing precision (see Materials and methods). We found that the firing precision of models

largely mirrored the spiking resonance preference. Specifically, somatic H models exhibited high synchronization (defined as VS >0.8) at 2–5 Hz whereas models with somatodendritic H were highly synchronized at the higher theta range of 4–9 Hz (*Figure 3A*). With H block, models with somatic H exhibited no significant reduction in synchronization at any input frequency (*Figure 3B*, left) whereas models with somatodendritic H showed a significant decrease in synchronization at 3–15 Hz (*Figure 3B*, right). The mean phase of firing was not examined in the subset of frequencies with low synchronization (VS <0.6) since the concept of mean phase is not informative when synchronization is poor. The phase of firing was defined so that 0° corresponds to the release from peak inhibition during each phase of modulated input (i.e., arrows in *Figure 1—figure supplement 3*). Accordingly, model responses at 3 Hz, 5 Hz, and 8 Hz – a representative sample of preferred frequency ranges for models across both distributions of H – showed that, in all cases, somatodendritic H models exhibited a phase advance compared to somatic H models (*Figure 3C*). Interestingly, with 8 Hz modulation, somatodendritic H models fired at 180° relative to release from inhibition. As described in the Discussion below, this is in line with in vivo data on theta phase-specific firing of O-LM cells (*Varga et al., 2012*). The differential theta frequency preference of models is further illustrated in the 3 Hz, 5 Hz and 8 Hz input modulation simulations by aligning all spikes within an example model's output with respect to the phase of inhibition (*Figure 3D*). The example model with somatic H (rank 26, *Figure 3D*, top row) exhibited tighter clustering of spikes at the same relative phase of input at 3 Hz compared to 8 Hz, whereas the example model with somatodendritic H (rank 109, *Figure 3D*, bottom row) displayed more synchronized spikes at 8 Hz compared to 3 Hz. Both classes of models could synchronize well at 5 Hz (*Figure 3A*, VS >0.8 for both $H_S$ and $H_{SD}$ at 5 Hz) although they differed in preferred phases (*Figure 3C*, middle and *Figure 3D*, middle).

Taken together, these results demonstrate that, depending on H distribution, O-LM cell models differ in a largely non-overlapping manner regarding which inhibitory synaptic input frequencies entrain and precisely recruit O-LM cell spiking. Somatic H models are entrained at low theta frequencies (2–5 Hz) whereas somatodendritic H models are entrained at high theta frequencies (4–9 Hz). Blockade of H results in impairment in both recruitment and precision for each class of models' respective preferred firing frequency ranges, with a more severe reduction seen in the somatodendritic H models' ability to fire in response to high theta modulated input.

## A balance of H and Kdrs conductances underlies the theta spike resonating ability of somatodendritic H models

Despite the reduction in spiking recruitment of somatodendritic H models, their broad theta spiking resonance peak was nevertheless maintained despite H block (*Figure 2C*, left, orange). In contrast, somatic H models did not exhibit the same broad theta spiking response, whether in control or with H block (*Figure 2B*). The question thus arose of what other conductances, aside from H, underlie the ability of somatodendritic H models to be entrained in the 4–9 Hz frequency range. To understand the differing spiking dynamics between somatic and somatodendritic H models and obtain clues as to candidate conductances, we first examined the differences in the time course of post-spike somatic $V_m$ trajectories. We found that, with input modulation, somatic H models exhibited a sharp depolarization immediately upon spike termination, whereas somatodendritic H models were held longer at potentials closer to the spike afterhyperpolarization (*Figure 4A*). This pointed to the role in somatodendritic H models of a slower, outward potassium current that would keep the membrane more hyperpolarized after each spike. In our previous work on building and analyzing the model database from which we extracted the models used here, we found co-regulatory balances only in somatodendritic H models involving H and two other potassium conductances (*Sekulić et al., 2014*). These consisted of the slow-delayed rectifier (Kdrs) and the A-type potassium (KA) conductances. We thus focused on Kdrs and KA as potential candidates underlying the differences in post-spike subthreshold voltages and the ability of somatodendritic H models to exhibit 4–9 Hz theta spiking resonance.

Furthermore, because each set of somatic and somatodendritic H models were chosen to have the same value of H maximum conductance, a candidate outward conductance underlying the somatodendritic H models' broad theta spiking resonance peak should necessarily have to take on consistently different values between somatic and somatodendritic H models. This is because none of the somatic H models exhibited a similar broad theta spiking resonance, despite possessing both candidate conductances (*Figure 2B*, left). Accordingly, the model parameters show that Kdrs takes on an

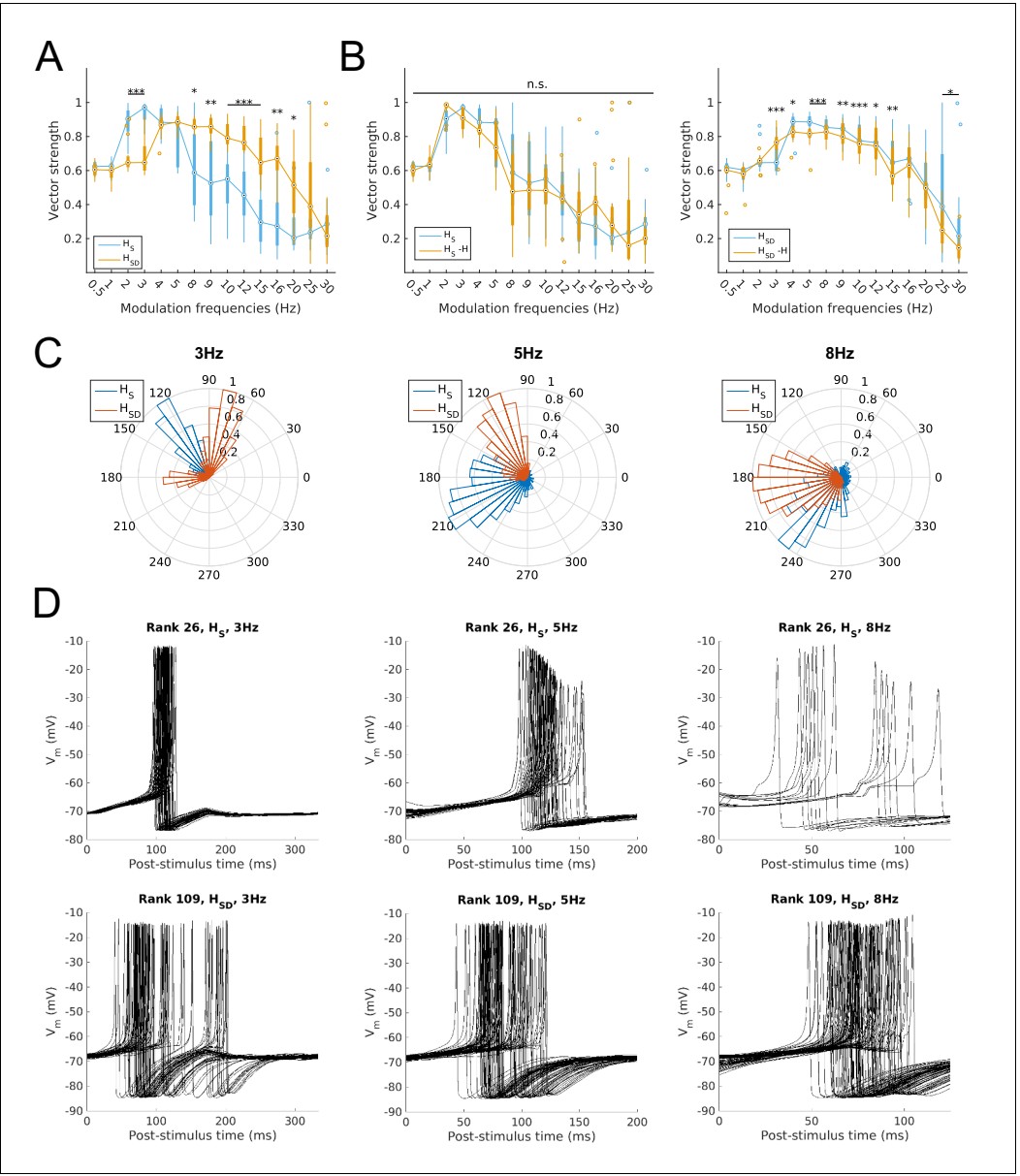

**Figure 3.** Firing precision and phase of models with modulated somatodendritic inputs. (**A**) Firing precision (vector strength) across models with somatic H (blue) and somatodendritic H (orange) in control (without H block), across all modulation frequencies ($F_{(1,15)}$ = 9.378, p<0.001, n = 16; Huynd-Feldt correction). Boxplot annotations as per *Figure 1G* legend. (**B**) Vector strength across models with somatic H ($H_S$, left) and somatodendritic H ($H_{SD}$, right), in control (blue) and with H blocked (orange), across all modulation frequencies. Statistical test used was two-way repeated measures ANOVA for the populations of $H_S$ and $H_{SD}$ models between all modulation frequencies crossed with H block condition ($H_S$: $F_{(1,15)}$ = 1.682, p=0.13, n = 16; $H_{SD}$: $F_{(1,15)}$ = 4.45, p=0.009, n = 16; Huynd-Feldt correction reported for both tests). Boxplot annotations as per *Figure 1G* legend. (**C**) Firing phase histograms for models with somatic H (blue) and somatodendritic H (orange) for modulation frequencies of 3 Hz (left), 5 Hz (middle), and 8 Hz (right). (**D**) Overlay of $V_m$ traces of all spikes for a sample somatic H model (top row) and somatodendritic H model (bottom row), cut and aligned with respect to the time of release from inhibition at 3 Hz (left), 5 Hz (middle), and 8 Hz (right).

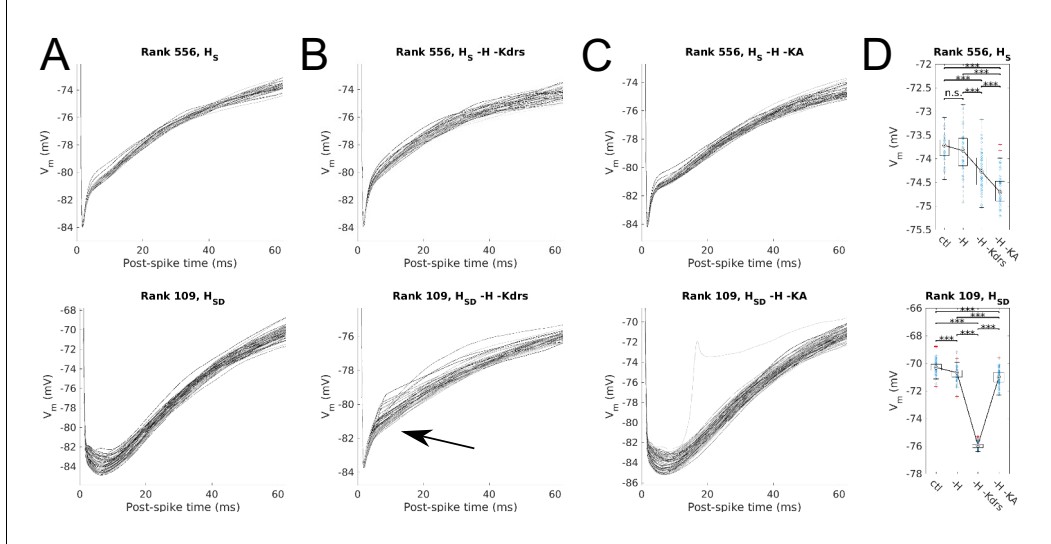

**Figure 4.** Post-spike $V_m$ trajectories at 8 Hz modulation for models with Kdrs and KA blocked. Overlays of the post-spike $V_m$ trajectories plotted by aligning the post-spike $V_m$ at spike peaks, for 8 Hz modulation simulations. Traces end halfway to the next 8 Hz theta cycle (62.5 ms) and are averaged for each model. One set of traces is shown per model, with each model's averaged post-spike $V_m$ trace given a different shade of grey to facilitate visualization. Cases shown are models with somatic H ($H_S$, top) and somatodendritic H ($H_{SD}$, bottom) in control (**A**), H and Kdrs blocked (**B**), and H and KA blocked (**C**). Arrow in (**B**) denotes where post-spike $V_m$ trajectories in the $H_{SD}$ model changes to resemble that of an $H_S$ model with H and Kdrs block (see main text). (**D**) The distribution of averaged $V_m$ values at the halfway point to the next theta cycle, i.e., the values shown at the end of each of the traces for the conditions shown, for the somatic H (top) and somatodendritic H (bottom) models. Statistical test used was 2-sample t-test with p<0.001 (***) and n.s. denoting not significant. n = 16 for each $H_S$ and $H_{SD}$ population.

The following figure supplement is available for figure 4:

**Figure supplement 1.** Effect of blocking H as well as Kdrs and/or KA on sub- and suprathreshold measures.

order of magnitude higher conductance density for all somatodendritic H models, compared to somatic H models, whereas KA takes on a comparable range of values both in somatic as well as somatodendritic H models (*Table 1*). We thus hypothesized that Kdrs in particular would be the most likely candidate for a conductance working in tandem with H to produce 4–9 Hz theta spiking resonance in somatodendritic H models.

Initial block of either Kdrs or KA alone led to uncontrolled firing in all O-LM models due to loss of the outward current that counterbalanced H, especially in the case of Kdrs. The high-frequency firing persisted even after excitatory synaptic input was entirely removed (not shown). These models could not be used in our study since they could not be meaningfully situated in a high-conductance state with synaptic inputs. Accordingly, we blocked either Kdrs or KA in models that already had H blocked. We found that in either case, there was an increase in membrane depolarization leading to heightened excitability with no input modulation. Thus, as with the H block simulations, we retuned the peak excitatory conductance on a per-model basis, resulting in a new set of excitatory synaptic parameters to ensure that prior to modulation of the inhibitory inputs, all models were situated in the same high-conductance state of ~2.5 Hz firing (*Figure 4—figure supplement 1*). We then simulated modulated inputs for all models and assessed once more the post-spike subthreshold dynamics. We found that blocking both H and Kdrs led to no overall change in post-spike $V_m$ trajectories for somatic H models, but changed the post-spike $V_m$ trajectories for somatodendritic H models such that they resembled that of somatic H models (*Figure 4B*, arrow). Thus, blocking both H and Kdrs turned a 'somatodendritic H type' model into a 'somatic H type' model. On the other hand, blocking H and KA did not change the post-spike $V_m$ trajectories for either somatic or

somatodendritic H models (*Figure 4C*). This suggested that Kdrs in particular plays a role together with H to provide the theta frequency spiking resonance features of somatodendritic H models.

To confirm that H and Kdrs together work to provide theta spiking resonance of somatodendritic H models in the high theta range, we calculated power ratios and rotation numbers for all models with H and Kdrs blocked. We found that, strikingly, the broad theta spiking resonance peak in somatodendritic H models was completely abolished with H and Kdrs block (*Figure 5A*, left). In fact, the spectral profile of somatodendritic H models with H and Kdrs block resembled that of somatic H models, with a peak power at 2–4 Hz, or the low theta range (*Figure 5A*, left, compare with *Figure 5C*, left). In other words, the spiking resonance peak in somatodendritic H models shifted from the high theta to low theta (2–4 Hz) ranges with H and Kdrs block, thus overlapping with the spectral profile of somatic H models. The rotation numbers were also significantly reduced for somatodendritic H models with H and Kdrs block (*Figure 5B*, left). In particular, the modulation frequencies for which somatodendritic H models could exhibit one spike per input cycle (i.e., rotation number of 1) shifted from 4–5 Hz in control to 2–3 Hz with H and Kdrs block, which is the identical range for somatic H models (*Figure 5B*, left, compare with *Figure 5D*, left). Somatic H models, on the other hand, did not exhibit any substantial changes in their spectral profile (power ratios) or rotation numbers in the theta range, except for a significant decrease in power ratio at 3 Hz (*Figure 5C*, left and *Figure 5D*, left).

We then analyzed power ratios and rotation numbers for all models with H and KA blocked and found that, for both somatic and somatodendritic H models, there was no broad effect of blocking H and KA on either power ratio, or rotation number, except for an increase in 3 Hz and 5 Hz power ratio for somatodendritic H models as well as a small decrease in 3–5 Hz rotation numbers for somatic H models (*Figure 5A*, right; *Figure 5B*, right; *Figure 5C*, right; *Figure 5D*, right). Crucially, the ability of somatodendritic H models to exhibit a peak in spiking power in the high theta range, i.e., above 5 Hz, was unaffected, unlike with Kdrs block (*Figure 5A*, right, compare with *Figure 5A*, left).

With regards to the firing precision measures used, we found that the changes to model output largely mirrored that of spiking recruitment measures. In particular, vector strength in the high theta range was significantly reduced in somatodendritic H models with H and Kdrs block, to the point where they could not phase-lock well to 5–15 Hz modulated inputs (VS <0.8, *Figure 6A*, left). Interestingly, there was significantly higher synchronization at 2 Hz with H and Kdrs block compared to the case of no block, with an overall synchronization profile resembling that of somatic H, with highly synchronized output at 2–4 Hz (VS >0.8, *Figure 6A*, orange, compare with *Figure 3B*, left). Disruption in high theta synchronization was further reflected in the firing phase histograms, where the tight clustering of spikes at 5 Hz and 8 Hz inputs was lost in somatodendritic H models with H and Kdrs block (*Figure 6B*, middle and right, compare with *Figure 3C*). This was also observed in less tightly clustered spikes in somatodendritic H models at 5 Hz and 8 Hz compared to control (*Figure 6C*, top row, compare with *Figure 3D*, bottom row). Although output at 3 Hz remained highly synchronized (*Figure 6A* and *Figure 6C*, top left), the phase of firing was delayed with respect to control (*Figure 6B*, left, compare orange with grey as well as with *Figure 3C*, left).

When we examined the outputs of somatodendritic H models with H and KA block, we found no change in phase-locking strength except at 5 Hz (*Figure 6A*, right). The phase of firing of somatodendritic H models with H and KA block overlapped with the case of H and Kdrs block (*Figure 6B*), and clustering of spikes was unaffected at 3 Hz, 5 Hz, and 8 Hz compared to no block (*Figure 6C*, bottom, compare with *Figure 3D*, bottom). Finally, for somatic H models, there was no change in vector strength with either H and Kdrs block nor with H and KA block (*Figure 6—figure supplement 1*). These results collectively show that H and Kdrs, but not KA, constitute a core set of conductances that underlie the ability of somatodendritic H models to be entrained precisely at high theta frequencies, and that blocking of these conductances effectively 'transforms' the spiking profiles of somatodendritic H models into that of somatic H models.

## cAMP modulation of h-channels results in enhanced spiking resonance and recruitment of spikes at theta frequencies

So far, we found that O-LM models exhibited a high degree of recruitment and firing precision at either low or high theta ranges depending on H distribution and a balance with Kdrs. Furthermore, blocking H resulted in a modest reduction in the preferred theta ranges for each H distribution, with

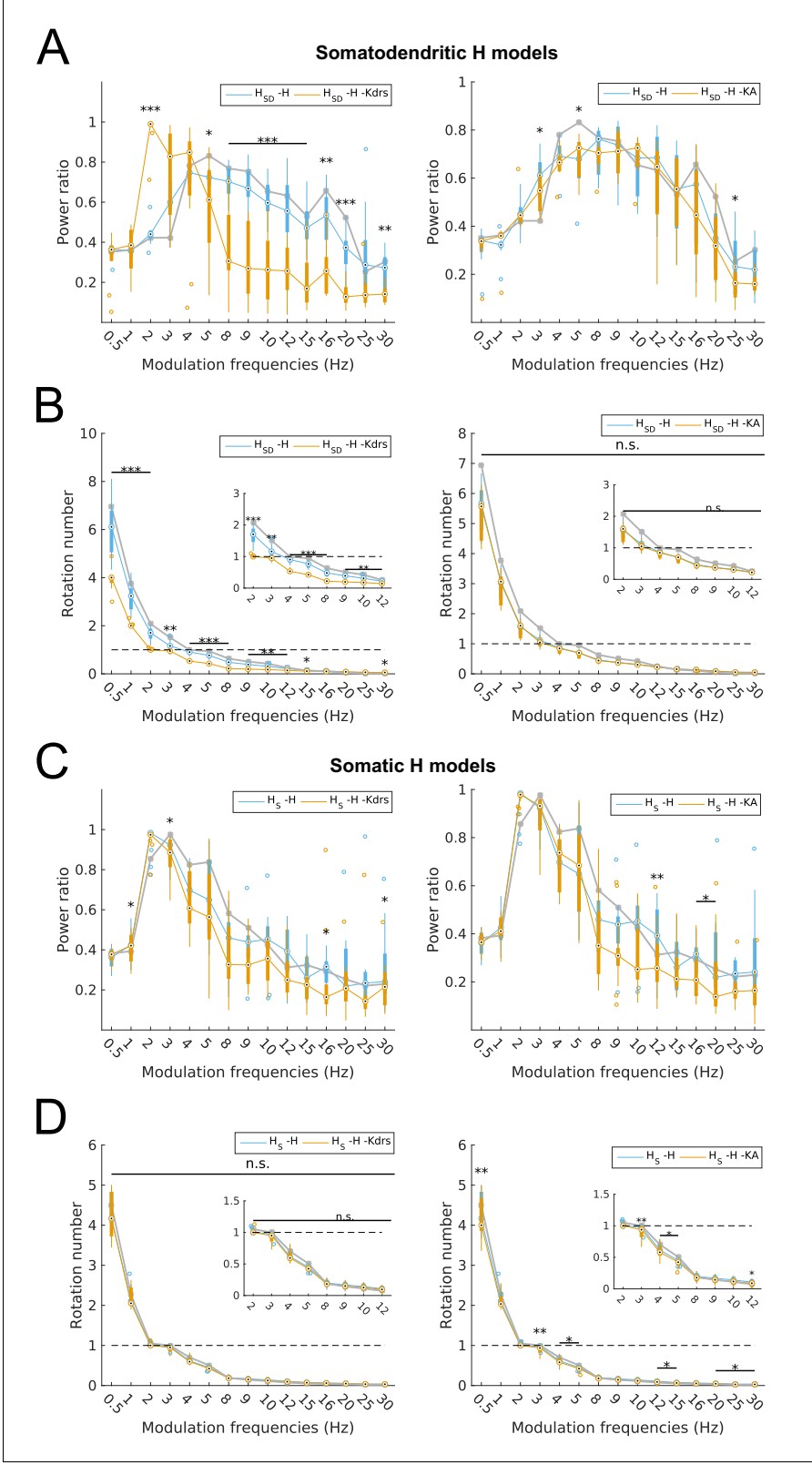

**Figure 5.** Changes to spiking recruitment in models with block of H and Kdrs or KA. Power ratios (**A**) and rotation numbers (**B**) for somatodendritic H ($H_{SD}$) models with either H and Kdrs blocked (left), or H and KA blocked (right). Power ratios (**C**) and rotation numbers (**D**) for somatic H ($H_S$) models with either H and Kdrs blocked (left), or H and KA blocked (right). In all cases, the control condition is H blocked only, without additional Kdrs and KA block. The

*Figure 5 continued on next page*

*Figure 5 continued*

grey line depicts median values of corresponding cases without H block. Insets in rotation number plots in (**B**) and (**D**) show zoomed portion in the theta range (2–12 Hz). Statistical test used was repeated measures ANOVA for the relevant populations between all modulation frequencies crossed with H block condition ((A) -H -Kdrs: $F_{(1,15)}$ = 14.44, p<0.001, n = 12; -H -KA: $F_{(1,15)}$ = 4.20, p<0.001, n = 8; (B) -H -Kdrs: $F_{(1,15)}$ = 21.20, p=0.001, n = 12; -H -KA: $F_{(1,15)}$ = 2.4768, p=0.1576 (n.s.), n = 8; (C) -H -Kdrs: $F_{(1,15)}$ = 2.81, p=0.009, n = 16; -H -KA: $F_{(1,15)}$ = 2.43, p=0.03, n = 16; (D) -H -Kdrs: $F_{(1,15)}$ = 0.33, p=0.59, n = 16; -H -KA: $F_{(1,15)}$ = 6.51, p=0.02, n = 16; Huynd-Feldt correction reported for all tests). Boxplot annotations as per *Figure 1G* legend.

complete abolishment of high theta recruitment for somatodendritic H models with H and Kdrs block. We next examined the converse, and asked whether *enhancement* of H could lead to *improved* theta spiking power. One of the signature features of HCN channels is the facilitation of activation of H current by direct binding by cyclic AMP (cAMP), which shifts the voltage dependency of activation ($V_{1/2}$) to depolarized potentials (*Biel et al., 2009*). This effect can be amplified by the elevation of intracellular cAMP levels via the binding of β-adrenergic receptors by the neurotransmitter noradrenaline (*Kupferman, 1980*). This was demonstrated in O-LM cells in vitro by bath application of noradrenaline, which resulted in increased H current at depolarized, but not hyperpolarized $V_m$, due to a shift in $V_{1/2}$ compatible with elevated intracellular cAMP (*Maccaferri and McBain, 1996*). Accordingly, we simulated the effects of modulation of H by cAMP to explore the effects of enhanced H on O-LM cell spiking activity at theta. We chose to simulate cAMP modulation of H by shifting $V_{1/2}$ in models by +5 mV, from the control case of –84 mV to –79 mV. The choice of +5 mV falls well within the range of shifts in $V_{1/2}$ previously reported in recombinant HCN channels with administration of cAMP (*Wainger et al., 2001*; *Baruscotti et al., 2015*). In addition, a previous study using dynamic clamp in O-LM cells used a $V_{1/2}$ of –75 mV in experiments examining the effects of a depolarized shift in activation of H in O-LM cell spiking (*Kispersky et al., 2012*). Thus, our choice of a +5 mV shift to –79 mV was conservative in comparison. In the below, we refer to this shift in $V_{1/2}$ as cAMP modulation.

We performed simulations for the full range of frequency-modulated synaptic inputs using models with cAMP modulation as per the above. No changes in the synaptic background activity was performed here as this was considered an in silico experimental manipulation. Also, physiological changes in intracellular cAMP are presumably not accompanied by rescaling of synaptic background activity in biological O-LM cells. We found that there was a shift in the peak spiking resonance for both somatic and somatodendritic H models (defined as frequencies with power ratio >0.6). For the former, the peak shifted from 2–5 Hz in control to 3–5 Hz with cAMP (*Figure 7A*, left). For somatodendritic H models, on the other hand, not only did the peak power ratio shift from 4–12 Hz to 8–10 Hz, but spiking power was suppressed for input frequencies outside of this 8–10 Hz range (*Figure 7B*, left). Examining the rotation numbers, somatic H models shifted the frequencies for which they could be recruited for a majority of input cycles (rotation number between 0.5 and 1) from 2–5 Hz to 3–5 Hz (*Figure 7A*, right). On the other hand, somatodendritic H models shifted their frequencies under which a spike was evoked for the majority of input cycles from 4–8 Hz to 8–12 Hz, coinciding with the shifted peak in power ratio (*Figure 7B*, right). The effectiveness by which somatodendritic H models could be recruited in the 8–10 Hz theta range under cAMP modulation can be readily seen in the intracellular $V_m$ recordings of the simulations and associated power spectral density (PSD) plots (*Figure 7C*).

With respect to firing precision, we found that somatic and somatodendritic H models with cAMP modulation, as with spiking recruitment, exhibited a concomitant shift in input frequencies for which they could be precisely recruited (VS >0.8). A shift from 2–5 Hz to 3–5 Hz was found in somatic H models with cAMP modulation (*Figure 8A*, left), and a shift from 4–10 Hz to 8–10 Hz was found in somatodendritic H models with cAMP modulation (*Figure 8A*, right). Suppression of phase-locking ability to frequencies outside 8–10 Hz was also found in somatodendritic H models, mirroring the reduction in spiking power outside this same range (*Figure 8A*, right). For frequencies with high phase locking, there was a phase advance in the mean phase of firing for somatic H models with 3 Hz but not 8 Hz or 10 Hz (*Figure 8B* top row), and for all three frequencies for somatodendritic H models (*Figure 8B*, bottom row). This was further reflected in the tight clustering of spikes for

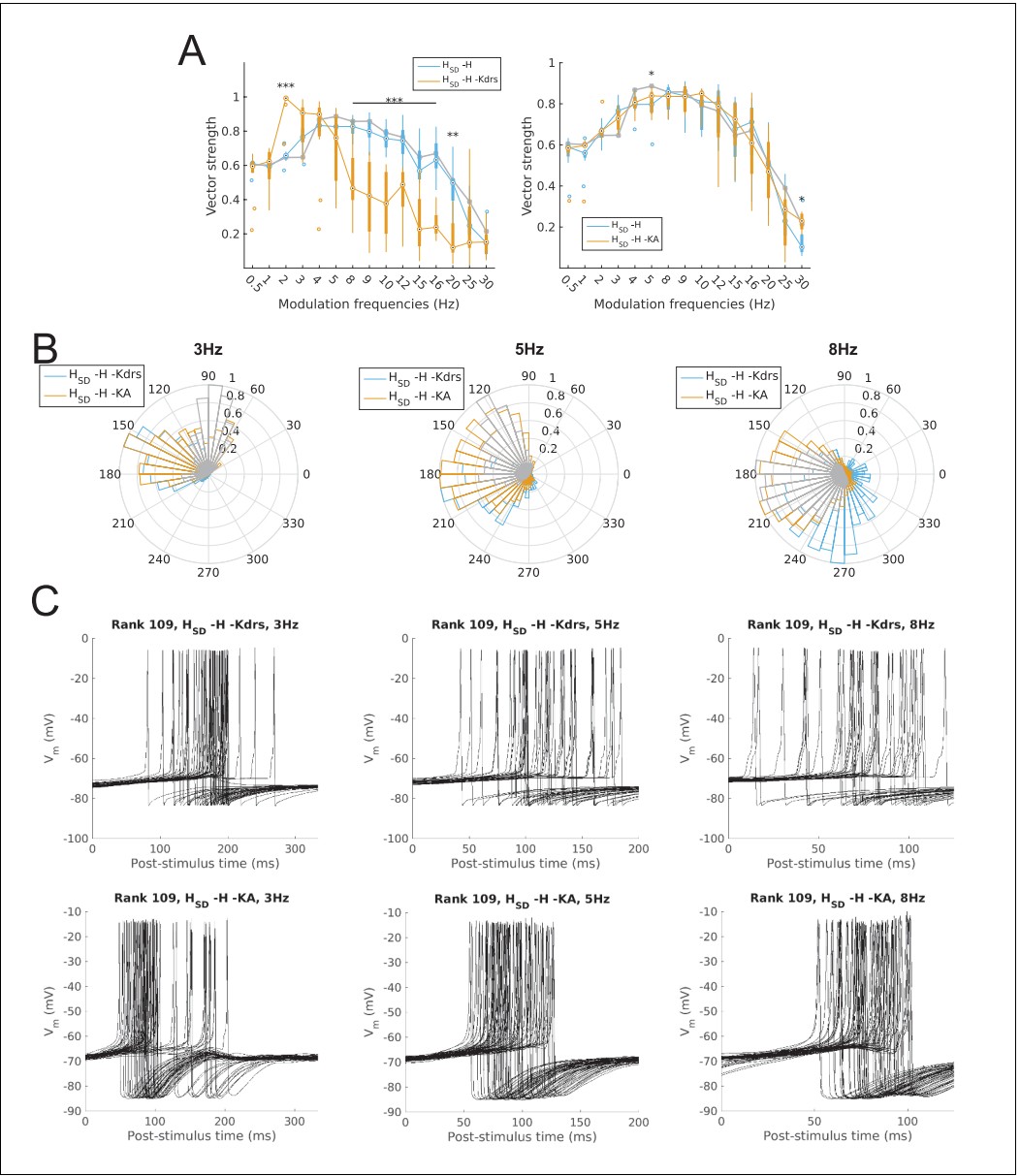

**Figure 6.** Changes to firing precision and phase in somatodendritic H models with block of H and Kdrs or KA. (**A**) Firing precision (vector strength) across somatodendritic H ($H_{SD}$) models, with H and Kdrs blocked (blue) and with H and KA blocked (orange) across all modulation frequencies ($F_{(1,15)}$ = 16.00, p<0.001, n = 8; Huynd-Feldt correction). Boxplot annotations as per *Figure 1G*. (**B**) Vector strength across $H_{SD}$ models with only H blocked (blue, -H) and with H and Kdrs blocked (left, orange, -H -Kdrs) and H and KA blocked (right, orange, -H -KA), across all modulation frequencies. The grey line depicts median values of corresponding cases without H block (control). Statistical test used was two-way repeated measures ANOVA test for the population of -H -Kdrs and -H -KA models between all modulation frequencies crossed with H and Kdrs/KA blocked condition (-H -Kdrs: $F_{(1,15)}$ = 18.739, p<0.001, n = 12; -H -KA: $F_{(1,15)}$ = 2.83, p=0.01, n = 8; Huynd-Feldt correction reported for both tests). Boxplot annotations as per *Figure 1G*. (**C**) Firing phase histograms for $H_{SD}$ models with H and Kdrs blocked (blue) and H and KA blocked (orange) for modulation frequencies of 3 Hz (left), 5 Hz (middle), and 8 Hz (right). In all cases, the control condition is H blocked only, without additional Kdrs and KA block. (**D**) Overlay of $V_m$ traces of all spikes for an example $H_{SD}$ model with H and Kdrs blocked (top row) and H and KA blocked (bottom row), cut and aligned with respect to the time of release from inhibition at 3 Hz (left), 5 Hz (middle), and 8 Hz (right).

The following figure supplement is available for figure 6:

**Figure supplement 1.** Changes to firing precision in somatic H models with block of H and Kdrs or KA.

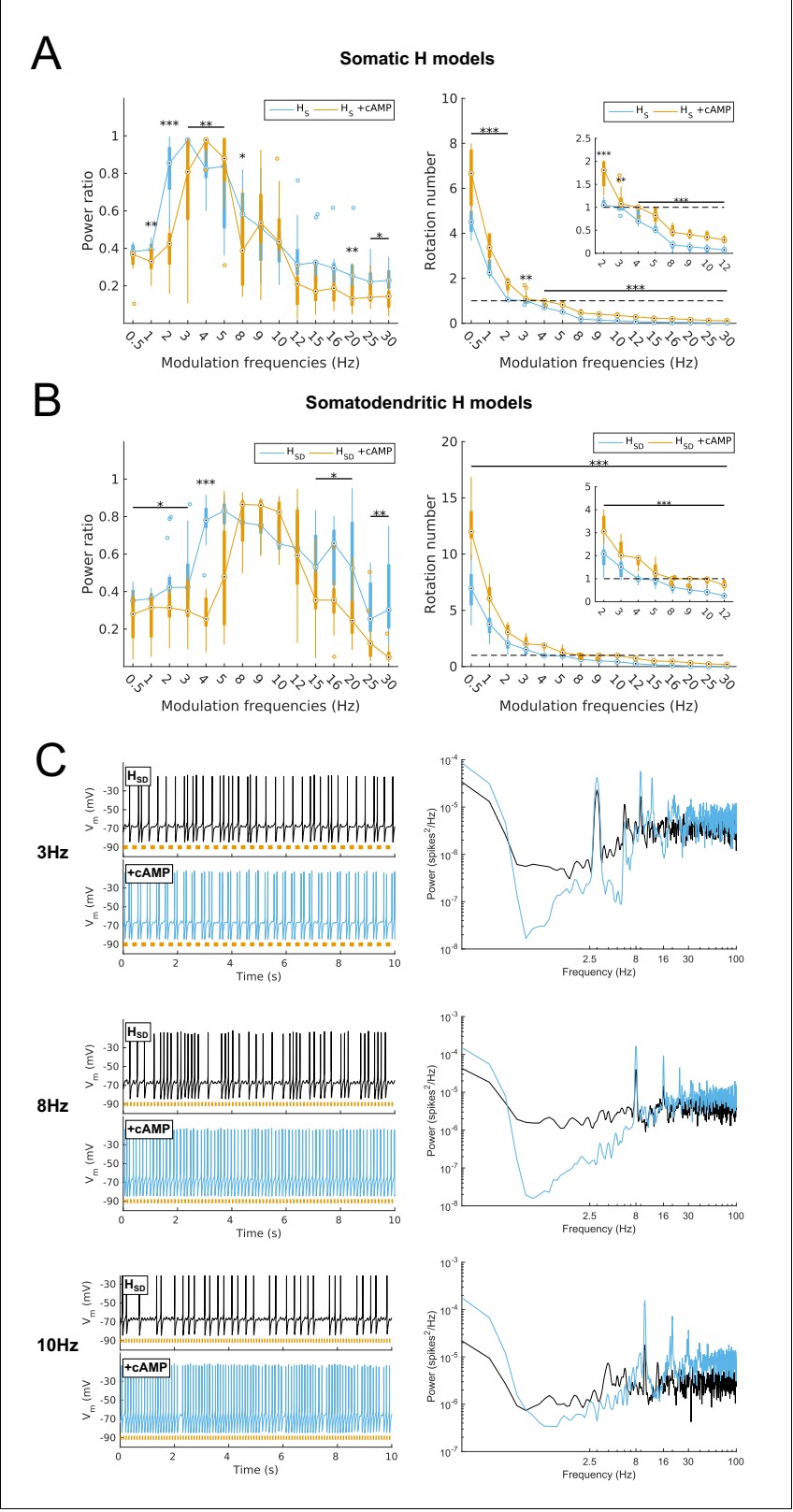

**Figure 7.** Effects on spiking resonance of simulating cAMP modulation of H channels. (**A**, **B**) Power ratios (left) and rotation numbers (right) under different modulation frequencies for somatic H models (B, $H_S$) and somatodendritic H models (C, $H_{SD}$) in control and cAMP ('+cAMP') conditions. Insets in rotation number plots show zoomed portion in the theta range (2–12 Hz). Statistical test used was repeated measures ANOVA for the populations of $H_S$

*Figure 7 continued on next page*

*Figure 7 continued*

and $H_{SD}$ models between all modulation frequencies crossed with cAMP condition (power ratios $H_S$: $F_{(1,15)}$ = 18.66, p<0.001, n = 16; $H_{SD}$: $F_{(1,15)}$ = 5.16, p=0.003, n = 16; rotation numbers $H_S$: $F_{(1,15)}$ = 79.23, p<0.001, n = 16; $H_{SD}$: $F_{(1,15)}$ = 59.52, p<0.001, n = 16; Huynd-Feldt correction reported for all tests). Boxplot annotations as per *Figure 1G* legend. (C) Somatic $V_m$ traces for an example model in control (black) and cAMP (blue) under various modulation conditions (top – no modulation; middle – 3 Hz modulation; bottom – 8 Hz modulation). With modulated inputs, orange bars at bottom denote phase of peak modulation at the specified frequency (see Materials and methods). Power spectrum density (PSD) plots shown to the right of each set of output traces.

somatic H models with 3 Hz, and for somatodendritic H models with 8 Hz and 10 Hz inputs (*Figure 8C*).

Overall, with cAMP modulation, models with differing H distributions still exhibited a division of preferred recruitment and firing precision responses into low and high theta, similar to that found in the control simulations. For somatic H models, this was in a slightly shifted low theta range at 3–5 Hz with all measures (power ratio, rotation number, and VS), compared to 2–4 Hz (power ratio) and 2–5 Hz (rotation number and VS) in control. For somatodendritic H models, on the other hand, recruitment and firing precision was seen in a narrower shift into the upper high theta range, with 8–10 Hz (power ratio and VS) and 8–12 Hz (rotation number) compared to 4–12 Hz (power ratio) and 4–9 Hz (rotation number and VS) in control. Therefore, cAMP modulation confers phase advance of firing in models, and narrowing of H recruitment and precision in the respective low vs high theta ranges depending on H distribution, with an additional enhancement in the high theta range for somatodendritic H models at 8–10 Hz.

## Discussion

In generating detailed, multi-compartment models representing different cell types, it is important to have ongoing, reciprocal interactions between experiment and computational modelling rather than a primary focus on producing models that best fit the data (*Sekulić and Skinner, 2017*). This is because the goal of multi-compartment modelling should not primarily be about determining the densities and distributions of different channel types, which we know are not fixed, but rather to understand the importance of these variables in shaping functional output. In general, the challenge we need to meet is how to link biophysical and cellular characteristics with brain function (*Gjorgjieva et al., 2016*).

The wide diversity yet critical contributions of different types of interneurons in brain function is apparent (*Kepecs and Fishell, 2014*). O-LM cells were one of the first clearly identifiable inhibitory cell types in the hippocampus and, with the development and use of sophisticated experimental techniques, we now appreciate its specific cellular characteristics as well as its importance in gating information flow (*Martina et al., 2000*; *Leão et al., 2012*). However, many unaddressed questions regarding their functional roles remain. In particular, although O-LM cells fire phase-locked to theta rhythms in vivo (*Klausberger et al., 2003*), the manner in which they are recruited during theta network activity is unclear.

In this paper we have performed the equivalent of biological experiments in silico, allowing us full control over, and access to, the experimental manipulations and variables of interest. In particular, we have used detailed multi-compartment models of O-LM cells to examine whether they fire preferentially at theta frequencies when driven by synaptic excitatory and inhibitory conductances. We situated our models within high-conductance, artificial in vivo-like states and modulated the inhibitory inputs at different frequencies, including theta (4–12 Hz). We found that our models were recruited to spike preferentially at theta frequencies with a recruitment precision and phasing that depended on whether h-channels were present in the soma only or also in the dendrites. Models with somatic only h-channels exhibited a low theta frequency spiking preference (2–5 Hz) whereas models with somatodendritic h-channels demonstrated a high theta frequency spiking preference (4–9 Hz). We furthermore found that the preferential high theta frequency spiking resonance depended on the presence of h-channels as well as slow delayed-rectifier potassium channels. Finally, we showed that the high theta preference of models with somatodendritic h-channels, but

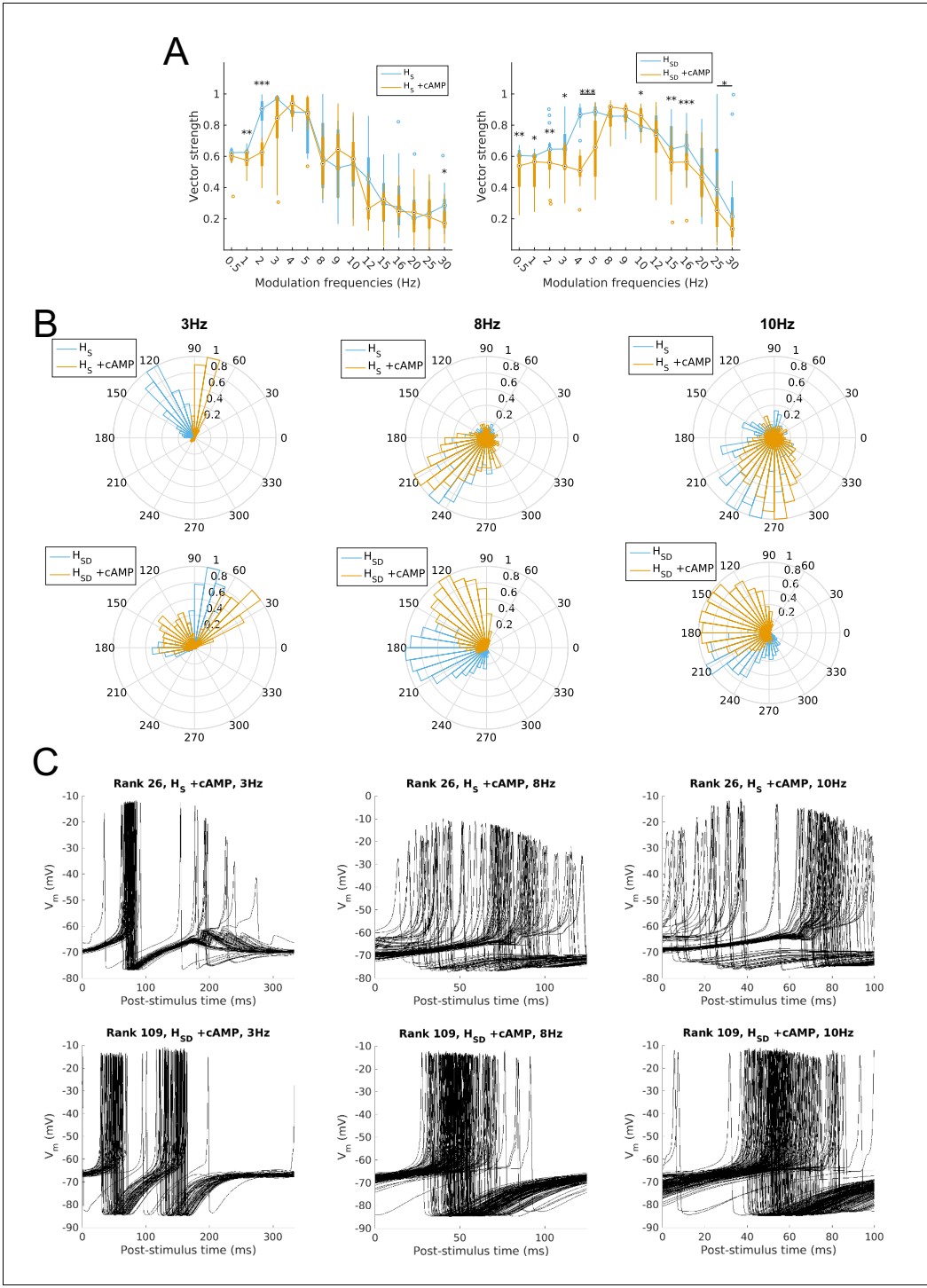

**Figure 8.** Firing precision and phase of models with cAMP modulation of H channels. (**A**) Vector strength across models with somatic H ($H_S$, left) and somatodendritic H ($H_{SD}$, right), in control (blue) and with cAMP (orange) conditions, across all modulation frequencies. Statistical test used was two-way repeated measures ANOVA test for the populations of $H_S$ and $H_{SD}$ models between all modulation frequencies crossed with H blocked condition ($H_S$: $F_{(1,15)}$ = 4.20, p=0.003, n = 16; $H_{SD}$: $F_{(1,15)}$ = 6.38, p<0.001, n = 16; Huynd-Feldt correction reported for both tests). Boxplot annotations as per **Figure 1G** legend. (**B**) Firing phase histograms for models under cAMP condition with somatic H (top) and somatodendritic H (bottom), for control (blue) and cAMP (orange) conditions, and modulation frequencies of 3 Hz (left), 8 Hz (middle), and 10 Hz (right). (**C**) Overlay of $V_m$ traces of all spikes for

*Figure 8 continued on next page*

*Figure 8 continued*

an example somatic H model with cAMP (top row) and somatodendritic H model with cAMP (bottom row), cut and aligned with respect to time of release from inhibition at 3 Hz (left), 8 Hz (middle), and 10 Hz (right).

not the low theta preference of models with somatic h-channels, could be shifted to higher theta frequencies (8–10 Hz) when modulation of h-channels by cyclic AMP (cAMP) was simulated.

The breadth of our results suggests that O-LM cells in vivo may be differentially and flexibly recruited by frequency-modulated synaptic inputs depending on intrinsic channel distributions and conductance densities. We note that it was possible to carry out this work because of the existence of our previously developed O-LM cell model databases with differential h-channel distributions (*Sekulić et al., 2014*, *2015*). Our dissection of channel mechanisms underlying theta spiking resonance in the models was further guided by the observed co-regulations between h-channels and two other potassium channels in our earlier modelling study (*Sekulić et al., 2014*).

## Implications and predictions

The findings in our work have implications for understanding how channel distributions and conductance co-regulations in O-LM cells may contribute to their spiking properties within in vivo-like contexts. Of particular note is the implication that a system of conductances, consisting of h-channels and slow-delayed rectifier potassium channels, may work together to endow subsets of O-LM cells to be recruited at different theta frequency ranges given a high-conductance state characterized by high synaptic bombardment and modulated inhibitory synaptic inputs. In *Kispersky et al. (2012)*, although post-spike afterhyperpolarization refractory dynamics were shown to play a role in O-LM spiking resonance, a particular channel was not identified as being responsible, whereas in our models, we could make a direct link and prediction to the role of slow delayed-rectifier potassium currents in helping shape spike resonance. Furthermore, this system of channels is flexible in that modulation of channel properties such as activation kinetics can affect the frequency tuning of preferred spiking. We demonstrated this in our models by simulating the effect of cAMP modulation on h-channel activation, which has been shown to shift the activation curve of h-channels ($V_{1/2}$) to more depolarized potentials (*Biel et al., 2009*). Modulation by cAMP via a modest +5 mV change in $V_{1/2}$ of h-channel activation shifted the theta spiking preference of the subset of models with somatodendritic h-channel distributions only to higher theta values (8–10 Hz). These results suggest that h-channels could be a modulatory target to exert flexibility in O-LM cell recruitment during network states; for instance, noradrenaline, which acts via elevation of cAMP, potentiates h-channels in O-LM cells (*Maccaferri and McBain, 1996*).

Although blocking h-channels alone lowered the spiking power in models with somatodendritic h-channels, it did not remove entirely the high theta firing preference of these models, whereas blocking both channel types completely abolished the high theta spiking preference, shifting the preference of firing to lower theta. Indeed, in this way we could 'convert' high theta-preferring models to low theta-preferring ones. The frequency ranges corresponding to 'low' and 'high' theta are dependent on what measures are used (power ratio, rotation number, or vector strength). Here we used rotation number – or average number of spikes per input cycle – as a measure of spiking recruitment since it is a more intuitively understandable and relatable measure to experimental data than power ratio. Furthermore, given that we only consider spike times as an output measure and not subthreshold fluctuations, the power spectrum only includes spiking information, which is also captured by rotation number. We did not consider models with rotation number greater than one to be regularly recruited because these cases in our models occurred with lower modulation frequencies and were primarily a result of the synaptic conductances having been chosen to produce 2.5 Hz baseline firing in the models. In other words, not enough modulation typically occured in lower frequency bands (<2 Hz) to affect the baseline firing, and thus the rotation number mainly reflected the baseline firing, not modulation. This is further reflected in the low synchronization index for these frequencies (see, for instance, *Figure 3A*). Thus, using the metric of between 50% and 100% recruitment across input cycles, low and high theta preferred responses of models can be delineated to 2–5 Hz and 4–9 Hz.

We predict that the presence of dendritic h-channels in O-LM cells may be an important factor in determining their spiking resonance preferences. Whether O-LM cells express dendritic h-channels is currently unknown. Previous work has found HCN2 channel expression in the somata of CA1 O-LM cells, but did not specifically examine dendritic expression (*Matt et al., 2011*). Dendritic h-channel expression has been demonstrated in other cell types; for instance, CA1 pyramidal neurons exhibit a non-uniform dendritic distribution of h-channels, with sixfold larger current densities in distal dendritic regions compared to proximal ones (*Magee, 1998*). In our previous work, we examined both uniform and non-uniform distributions of dendritic h-channels and found that disparate distributions can nevertheless reproduce somatic O-LM cell responses so long as total membrane h-channel conductance is maintained across models (*Sekulić et al., 2015*). These findings were similar to those observed in cerebellar Purkinje inhibitory cells, where although experimental recordings found uniform dendritic h-channel expression, subsequent multi-compartment computational models with either uniform or non-uniform distributions but similar total h-channel conductance could account for the data equally well (*Angelo et al., 2007*). Our present results suggest that determining dendritic expression of HCN channel subtypes on O-LM cells, e.g., through immunohistochemical studies, is needed to shed light on functional properties of O-LM cells.

Given the minimal overlap of theta preference of our models with somatic h-channels versus somatodendritic h-channels, with the former preferring low and the latter high theta, it is suggestive to make a link between these frequency preferences in our models and the two types of theta rhythms observed in rodents (*Buzsáki, 2002*). We note, however, that the link we make is with the parsing of two different theta frequency ranges, and not the exact frequency values in these ranges per se. Specifically, O-LM models with somatodendritic h-channels exhibit spiking preference in higher (Type 1-like) theta range whereas models with somatic h-channels fire in the lower (Type 2-like) theta range. Type 1 theta originates in dorsal hippocampus and has been associated with spatial memory and cognitive function whereas Type 2 theta seems to be involved in emotional processing and may originate in ventral hippocampus (*Strange et al., 2014*; *Ciocchi et al., 2015*; *Patel et al., 2012*). Recent work has linked different behavioural states such as fear and anxiety with theta and intra-hippocampal circuitry (*Engin et al., 2016*). Furthermore, activation of ventral O-LM cells was found to promote Type 2 theta oscillations in ventral hippocampus (*Leão et al., 2015*). Taken together with our results, these factors suggest that, depending on dendritic distributions of h-channels and co-regulation with the slow-delayed rectifier channel, O-LM cells may be 'tuned' to be Type 1 theta or Type 2 theta spiking resonators, and thus that this system of channels may constitute an intrinsic 'switch' that varies depending on location along the dorsoventral axis.

Our model explorations focused on dendritic distributions of h-channels and not on the time constant of activation of h-channels per se. It is possible that comparable results may be obtained in our models if we kept h-channel distributions fixed (e.g., localized only to the soma) but varied the time constant of h-channel activation and thus the type of HCN subunit represented (faster kinetics due to HCN1 vs slower due to HCN2, for instance). Precedence for this possibility can be found in differences in the time constants of h-channels in Layer II stellate cells in medial entorhinal cortex along the dorsoventral axis (*Giocomo and Hasselmo, 2008*). Additionally, in CA1, a gradient of HCN1 to HCN2 was observed in pyramidal neurons across the dorsoventral axis, with higher ratios of HCN1-to-HCN2 in ventral pyramidal neurons and higher HCN2-to-HCN1 in dorsal pyramidal neurons (*Dougherty et al., 2013*). This resulted in ventral pyramidal cells being more excitable than dorsal pyramidal cells by virtue of a more depolarized $V_{1/2}$ of h-channels and a greater h-channel density compared to dorsal pyramidal cells. Interestingly, our work would predict that a reverse gradient of HCN1-to-HCN2 in dorsal vs. ventral O-LM cells exist, given that the models exhibiting high theta spiking preferences – putatively situated in dorsal hippocampus where the high (Type 1) theta is more prominent – exhibited h-channel characteristics resembling the ventral pyramidal neurons that express high HCN1 densities.

Although our simulations represented only a simplified in vivo-like context, we find a general congruence between our results and the hippocampal in vivo literature. For instance, O-LM cells fire at the trough of hippocampal theta frequencies in vivo (*Varga et al., 2012*) as measured in the pyramidale layer, a 180° phase delay from inhibitory inputs received from the medial septum-diagonal band of Broca, or MS-DBB (*Borhegyi et al., 2004*). Phase-specificity in interneuronal firing may be important for mediating the effects of inhibition during theta rhythms, as seen in experiments demonstrating the role of theta trough-preferring SOM+ cells, which include O-LM cells, in modulating burst

firing of CA1 pyramidal cells (*Royer et al., 2012*). Given that the modulated inhibitory inputs in our work could correspond to medial septal inputs, the resulting 180° average phase delay with somato-dendritic h-channel models would then correspond to the phase of firing of O-LM cells observed in vivo. Therefore, our results suggest that, given dendritic synaptic inputs, O-LM cells require somato-dendritic h-channel expression to be precisely recruited during the trough of hippocampal theta activity as observed in vivo.

## Limitations and future work

Although we use a limited in vivo representation with 'random' synaptic distribution over the dendritic tree – in other words, a simplified, agnostic approach – we consider it advantageous over recreating in vivo states from in vitro settings using dynamic clamp, as done by *Kispersky et al. (2012)*, since synaptic inputs do not need to be only somatically located when using multi-compartment models. That is, our models allowed us to delve further in examining O-LM cell contributions to theta rhythms. Indeed, we found substantial differences in cell recruitment depending on whether synapses were spread across the dendritic tree, compared to localization in the soma only. Furthermore, we can fully observe the biophysical densities and distributions of channels and parameters in our models, unlike the case in biological cells, and can control them to examine differences in h-channel distributions. However, our O-LM cells are of course model representations and not biological cells, and so it is essential that there be ongoing iterations with model and experiment to examine our model predictions and generate more detailed data on membrane properties that can then be fed back into the models. This will especially be relevant once more information becomes available on the types and locations of synaptic inputs onto subcellular domains of O-LM cells.

A limitation of the study relates to the generic nature of artificial synaptic inputs used, as well as the 'on/off' or square pulse nature of modulation as opposed to using a sinusoidal wave. Given the lack of information regarding available types, locations, and timing of specific synaptic inputs that could be incorporated into our models, we decided to probe our O-LM cell model responses to frequency-modulated inputs in a general fashion. Nevertheless, intracellular recordings of PV+ MS-DBB cells that are presumed to synapse onto O-LM cells show that they burst phase locked to theta with a burst length ranging from 40° to 160° (*Borhegyi et al., 2004*), which resembles more closely the discrete transition between periods of increased and decreased inhibitory synaptic rates as implemented here, compared to a smoothly varying sine wave that is more characteristic of filtered LFP recordings. In more realistic in vivo contexts, however, it will likely be the case that fine-tuned inputs – e.g. from inhibitory or excitatory MS-DBB cells (*Garrett et al., 2014*; *Fuhrmann et al., 2015*), local CA1 IS-3 (VIP$^+$) inhibitory cells (*Tyan et al., 2014*), cholinergic afferents (*Lawrence et al., 2006a*), etc. – may interact with particular channels and even specific subcellular compartments to differentially recruit O-LM cells depending on behavioural context. Future studies can explore specific synaptic distributions that encompass what is known about spatiotemporal input distributions.

We used two different types of measures in assessing spiking recruitment, power ratio and rotation number, following that done by others (*Kispersky et al., 2012*; *Lawrence et al., 2006a*). What the ideal measure to use for determining functional activity is debatable. However, given our study design of examining firing preferences in theta network activity regimes, the rotation number can be more directly related to phasic, oscillatory network activity than power ratio, hence our emphasis on its use in the interpretation of our results and delineating high and low theta ranges. However, using the power ratio as the measure of spiking recruitment leads to a similar interpretation of results as does rotation number. Furthermore, a related issue is how much recruitment of O-LM cells occurs during in vivo theta rhythms. Previous work showing intracellular O-LM cell spiking during theta activity demonstrates recruitment during nearly each phase of theta (*Figure 3* in *Klausberger et al. (2003)*; *Figure 2* in *Varga et al., 2012*). However, larger datasets of O-LM cell firing during in vivo theta activity would be needed to further verify this recruitment. We further note that since our work is not a network model and hence does not generate network-level theta, phasing of inputs in our models is rather defined as starting from the release from the peak of the modulated inhibition, i.e., invoking post-inhibitory rebound mechanisms. Although these details of theta phasing matter, there is not yet a clear consistency or rationale between network theta models and experiments.

## Concluding remarks

It is essential that there be an ongoing dialogue between modelling and experiment to ensure that the advantages, challenges and limitations of both can be fully appreciated, and so that our understanding of different inhibitory cell types can move forward (e.g., see *Mendonça et al., 2016*).

Much more is known about hippocampal pyramidal cells than interneurons and the range of work there, including the demonstration of non-uniform distribution of h-channels, has given rise to the idea of 'functional maps' in a single neuron given the specifics of dendritic ion channels (*Narayanan and Johnston, 2012*). For instance, computational investigations using multi-compartment models have suggested that, in CA1 pyramidal neurons, interactions between h-channels and TASK-like potassium leak channels could explain paradoxical findings such as membrane depolarization after h-channel block (*Migliore and Migliore, 2012*). Other modelling work with CA1 pyramidal neurons has predicted that A-type potassium channels modulate the efficacy of the h-current balance between conductance and current, which exhibit counteracting effects of decreasing input resistance and depolarizing the membrane potential, respectively (*Mishra and Narayanan, 2015*). It is thus reasonable to also consider dendritic functional maps for different inhibitory cell types, and specifically O-LM cells which, to our knowledge, has not been done to date. In our work, we show that h-current interacts with slow delayed-rectifier potassium currents, but not A-type currents, to endow spiking resonance in the upper theta range to models, but only when h-channels are distributed in the dendrites. This may be a way to allow different subsets of O-LM cells to be recruited differentially depending on behavioural context, e.g., spatial coding via Type 1 theta versus emotional processing via Type 2 theta, which thus require different intrinsic spiking resonant preferences that may arise from dendritic distributions of h-channels as shown in the present work. At minimum, our work here suggests that it is inappropriate to assume that particular functions may always be attributable to one individual channel type. Rather, multiple channels, such as h-channels and slow-delayed rectifier potassium channels, can collectively contribute to bringing about functional properties, such as preferred spike frequency resonances.

In conclusion, while there are many directions that one can follow, the most immediate outcome in the context of the present work would be immunohistochemical labelling of HCN distributions in O-LM cell compartments. The further development of O-LM cell models using experimental data in which morphology, passive and active electrophysiological recordings are available from the same cell is another clear next step (*Sekulić et al., 2015*). In addition, it will be important to perform mathematical analyses to dissect out the dynamical interactions that give rise to this theta frequency preferred spiking. In particular, how best to take advantage of resonance analyses (e.g., *Rotstein and Nadim, 2014*) needs to be determined since our models here indicate that an expansion beyond single compartment models would be required as well as reductions that maintain dendritic aspects of integration and function.

# Materials and methods

## Extraction and preparation of models from database

The O-LM cell models used in this work are based on a database of multi-compartment models that each include nine voltage-gated ionic currents (*Sekulić et al., 2014*). The model parameters of relevance to the present work are the maximum conductance densities, which we refer to here using the following abbreviations: Nad and Nas respectively for the dendritic and somatic transient sodium channel densities, Kdrf and Kdrs respectively for the fast and slow delayed rectifier potassium channel densities, KA for the A-type potassium channel density, CaT and CaL respectively for the L- and T-type calcium channel densities, KCa for the calcium-activated potassium channel densities, H for the hyperpolarization-activated mixed cation channel density, and M for the Kv7/KCNQ/M channel density. Parameter values for all models used are provided in *Table 1*.

A total of 32 multi-compartment models were extracted from our previously developed database, evenly split between those with somatic H conductance only and somatodendritic H conductance, the latter models all having H uniformly spread along the dendrites (*Sekulić et al., 2014*, *2015*). Since there were two morphologies of O-LM cells used in the database (morphology 1, surface area 16,193.6 $\mu m^2$; morphology 2, surface area 9,980.1 $\mu m^2$), models with each of the two dendritic H distributions were chosen so that there would be an equal number of models with each

morphology (*Table 1*). To ensure that differences in maximum conductance densities of H would not confound subsequent spiking output, we selected models with identical H maximum conductance densities within each group of models with the two different H distributions. Furthermore, the time constant of activation of H, as well as the passive membrane properties of each model, were optimized on a per-model basis using the current clamp data as described in *Figure 5* of *Sekulić et al. (2015)*. This was done to better match the activation kinetics of H, since we had previously found that the top-ranked models from our original database showed insufficient h-channel-dependent sag amounts relative to the experimental traces (*Sekulić et al., 2015*). Our adjusted models still exhibited good matches to O-LM cell electrophysiological characteristics (e.g., see *Figure 1—figure supplement 1* for four examples, and *Table 2* for passive properties). Also see *Sekulić et al. (2014)*, Supplementary Tables S1 and S2 for the electrophysiological measurements extracted from the experimental data. After adjusting to specifically fit the sag response, the 32 O-LM cell models were deemed to be appropriate representations of O-LM cells as their features fell within the experimental dataset.

All of our simulations were performed in the NEURON simulation environment (*Hines and Carnevale, 2001*; RRID:SCR_005393) using the CVODE adaptive time step integration method. We set the absolute error tolerance to be $1 \times 10^{-6}$ based on trial simulations. Our selection criterion was to lower the error tolerance until we ensured that spike times and shapes (as determined by V-$^{dV}/_{dt}$ plots) did not change. Simulations were executed on the SciNet high-performance computing cluster (*Loken et al., 2010*). The duration of all simulations was set to 30s of simulated time for somatic inputs, and 20s for somatodendritic inputs. The latter were set to a shorter time due to the increased computational resources needed to simulate all dendritic synaptic input processes. Simulation run times were set to be as long as possible yet still allow completion within the 48 hr runtime limit of SciNet. Occasionally some models did not complete simulations within the time limit; they were not included in the analyses for those respective results and this is reflected in the reported *n* values in the figure captions.

## Modelling of synaptic inputs

To address how the models would respond to frequency-modulated synaptic inputs within an in vivo-like context, we first situated them within a high-conductance state (*Destexhe, 2007*; *Destexhe et al., 2003*). This is a state of activity where continual synaptic bombardment produces a depolarized membrane potential close to threshold, a marked increase in the ratio of membrane synaptic conductance to somatic leak conductance, as well as a decrease in input resistance (*Destexhe and Paré, 1999*). We note that characteristics of high conductance states have only been directly measured in neocortical pyramidal cells (*Destexhe and Paré, 1999*). Estimated synaptic parameters for modeling and subsequent insights have also only been performed in neocortical pyramidal cells (see review of *Destexhe et al., 2003*). Previous work by *Kispersky et al. (2012)* have adapted these studies to situate O-LM cells in a high-conductance state using dynamic clamp. They used Poisson-based excitatory and inhibitory synaptic rates of, respectively, 500 Hz and 1000 Hz (*Kispersky et al., 2012*). For our present work, we adopted parameter specifics used by *Kispersky et al. (2012)* and used the same synaptic rates for the case of somatic synaptic input, with one excitatory and one inhibitory process in the soma. We note that O-LM cells may receive predominantly kainate-based excitation (*Goldin et al., 2007*; *Kispersky et al., 2012*) and appear to express GABA$_A$ receptors based on the types of interneurons currently known to inhibit them (*Tyan et al., 2014*). Thus, it was reasonable to set the kinetics of the synapses to be identical for excitatory and inhibitory inputs due to the similarity between GABA$_A$ and kainate receptor kinetics. Both classes of synapses were accordingly modelled using a sum of exponentials with $\tau_{rise} = 0.5$ms and $\tau_{decay} = 6.8$ms. Synaptic reversal potentials were set to $E_{exc} = 0$ mV and $E_{inh} = -80$ mV, as per *Kispersky et al. (2012)*.

For the case of somatodendritic synaptic inputs we chose to simply extend the somatic input case to avoid making specific choices about synaptic numbers, release sites, correlations, etc., without having particular experimental constraints available. We did this by inserting a pair of excitatory and inhibitory inputs at multiple locations distributed across the somatodendritic tree. We inserted synapses in the middle of every fourth segment of the model morphology resulting in 21 input locations for morphology 1 (*Figure 1B*, left) and 19 locations for morphology 2 (*Figure 1B*, right). Excitatory and inhibitory peak synaptic conductance parameters were chosen that resulted in models with

approximately 2.5 Hz spiking output and ~2 mV standard deviation of $V_m$ subthreshold fluctuations. Subthreshold fluctuations were determined by cutting spikes out of the model output traces and calculating means and standard deviations of the resulting voltage waveforms. These output characteristics were chosen to reproduce the dynamic clamp experimental protocol performed by (*Kispersky et al., 2012*), and to ensure a high-conductance state characterized by fluctuating membrane potentials situated near threshold (*Destexhe et al., 2003*). Due to the different ion channel conductance densities in each of the 32 models (*Table 1*), a parameter search for synaptic conductances had to be performed for each model. Thus, each model had a different combination of excitatory and inhibitory conductances that resulted in appropriate baseline characteristics (*Figure 1—figure supplement 2*). If multiple synaptic conductance parameters resulted in appropriate frequency of output, we selected the combination that maximized the standard deviation of subthreshold $V_m$ fluctuations since it was not always possible to obtain ~2 mV voltage fluctuations (see *Figure 1E*).

For the additional control condition of replacing H with a leak channel instead of blocking it, a voltage-independent 'artificial leak' (or 'H leak') conductance was inserted. It carries the same reversal potential as the H channel model but with a different per-model conductance density. The conductance density was fitted by taking as a baseline each model's H maximum conductance parameter, then scaling it by the value of the H steady state activation function at the mean of the subthreshold $V_m$ fluctuations in the control case with H intact. This still provided more depolarizing current for most models, and a parameter search for reducing the H leak conductance was performed to result in baseline 2.5 Hz firing prior to modulation.

After determining the combination of synaptic conductances required to maintain baseline activity for each model, modulation simulations were performed. These consisted of adjusting the inhibitory synaptic rate (1000 Hz) by ±40%, with a higher average Poisson rate during 'peak' periods of the modulated frequency and a lower rate during the 'trough' periods (*Figure 1—figure supplement 3*). Implementation details consisted of inserting instances of the *NetStim* class into each synaptic input location in NEURON, with one *NetStim* object for the excitatory input process, and either one *NetStim* for the inhibitory input process in the case of no input modulation, or two *NetStim* objects in the case of modulation. For the latter, one *NetStim* object was active during the first half of the input cycle, representing the 'peak' of inhibition and with an increase in the average inhibitory synaptic event, whereas the second *NetStim* was active during the second half, representing the 'trough' of inhibition and with a decrease in the average rate. The modulated frequency ranges used in the simulations were 0.5, 1, 2, 3, 4, 5, 8, 9, 10, 12, 15, 16, 20, 25, and 30 Hz, with finer granularity in the theta (4–12 Hz) range compared to higher frequencies since previous studies have shown that O-LM cells have preferential spiking below gamma frequencies (*Pike et al., 2000*). The reference model code can be found on ModelDB (accession number 182797). The implementation of the synaptic modulation, as well as full parameters for the 32 models used here, are included in the supplementary data online.

Input resistance for the high-conductance state was computed and compared to input resistance without synaptic inputs, in both cases using a –25 pA current step injection simulation. Input resistance was found to be at least half as large in the high-conductance, compared to no synaptic input case.

## Metrics for assessing spiking output

Model output analysis and statistical testing was performed using MATLAB (RRID:SCR_001622). The model spiking output (recorded from somatic $V_m$ in the simulations) were converted into binned binary trains, with 0 representing no spike and 1 representing a spike. The power spectrum density (PSD) of the binary trains were computed using the *pwelch* function in MATLAB, and the *power ratio*, i.e., ratio of the PSD at the modulated frequency to the value at 0 Hz was computed. Recruitment of spiking across modulatory phases was also measured using the *rotation number*, or average number of spikes per input cycle, across all input cycles in a model simulation (*Lawrence et al., 2006a*).

For the subsequent measures, the input stimulus was considered to be the release of the peak inhibitory amount at the start of each trough phase of stimulation, when depolarizing currents – both intrinsic and synaptic – could then drive the model to spiking threshold (e.g., arrows in *Figure 1—figure supplement 3*). The ability of the models to exhibit phase-locked firing with respect

to the modulatory input was assessed using the vector strength (VS), or synchronization index (*Mardia and Jupp, 2000*), as follows. The relative time of each spike to the release of the most recent phase of inhibition was calculated as $t_{\text{diff}}$, resulting in a set of relative spike times in the range of $[0,T]$, with $T$ being the period of the modulatory input. The vector strength was then calculated as, $VS = \sqrt{\left(\sum_i \cos(\theta_i)\right)^2 + \left(\sum_i \sin(\theta_i)\right)^2}/N$, where $\theta_i = 2\pi(t_i \bmod \mathrm{T})/\mathrm{T}$, is the phase of spike $i$ in radians and $N$ is the total number of spikes in the output trace. VS ranges from $[0,1]$ with 1 indicating perfectly synchronized spikes.

## Statistical analyses

For statistical tests between two populations of models consisting of a single measurement, an Anderson-Darling test was first performed on both populations to test for normality (*Stephens, 1974*). If the test rejected the null hypothesis for normality, the Wilcoxon rank sum test was then performed to test whether the two populations arose from the same distribution; otherwise, a paired-sample *t*-test was performed. For tests between two populations of models with repeated measures (e.g., different modulation frequencies, different channel blocking conditions, or cAMP modulation conditions) a two-way repeated measures ANOVA (rmANOVA) test was performed with Tukey's post-hoc tests. To reduce the type I error rate due to violations of the sphericity assumption, Huynh-Feldt corrections were performed on the resulting $F$ and $p$ values (*Huynh and Feldt, 1976*).

## Acknowledgements

This work was supported by NSERC of Canada, an Ontario Graduate Scholarship, and the SciNet HPC Consortium. The authors would like to thank V Barkley, SA Campbell, A Guet-McCreight, JJ Lawrence, S Mikulovic, and W Nicola for their helpful feedback on the manuscript.

## Additional information

### Competing interests

FKS: Reviewing editor, *eLife*. The other author declares that no competing interests exist.

### Funding

| Funder | Grant reference number | Author |
|---|---|---|
| Ontario Graduate Scholarship | Graduate Student Award | Vladislav Sekulić |
| Natural Sciences and Engineering Research Council of Canada | Discovery Grant,RGPIN 2016-06182,RGPIN 203700-11 | Frances K Skinner |
| SciNet High Performance Consortium | SciNet HPC Consortium | Frances K Skinner |

The funders had no role in study design, data collection and interpretation, or the decision to submit the work for publication.

### Author contributions

VS, Conceptualization, Data curation, Software, Formal analysis, Investigation, Methodology, Writing—original draft, Writing—review and editing; FKS, Conceptualization, Resources, Supervision, Funding acquisition, Validation, Writing—original draft, Project administration, Writing—review and editing

### Author ORCIDs

Vladislav Sekulić, http://orcid.org/0000-0001-5000-389X
Frances K Skinner, http://orcid.org/0000-0001-7819-4202

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
