## [Decision Letter]

Thank you for submitting your article "Computational models of O-LM cells are recruited by low or high theta inputs depending on h-channel distributions" for consideration by *eLife*. Your article has been reviewed by two peer reviewers, and the evaluation has been overseen by a Reviewing Editor and Eve Marder as the Senior Editor. The following individual involved in review of your submission has agreed to reveal his identity: Ivan Soltesz (Reviewer #2).

The reviewers have discussed the reviews with one another and the Reviewing Editor has drafted this decision to help you prepare a revised submission.

Summary:

Overall, this is an important and carefully performed study, which examines the role of H-currents in rhythmic firing of virtual OLM interneurons during imposed synaptic activity patterns designed to simulate in vivo activity. The study leverages multicompartmental OLM cell models developed from reconstructed neurons with distinct somatic and dendritic compartments. The strength of the approach is in using models in which different combinations of channel distributions were all selected to fit the OLM firing properties, as is likely to occur in biology. Using this powerful approach the authors show that, in contrast to a previous study that modeled an in-vivo like synaptic conductance state with exclusively somatic synaptic inputs and somatic h-conductance, explicit modeling of dendritic compartments that receive excitatory and inhibitory synaptic inputs and express h-channels enabled O-LM interneurons in CA1 to synchronize their spike frequency to the modulation frequency of a population of inhibitory synaptic inputs.

Essential revisions:

1) The role of H-currents and (M-type) potassium channels has been explored in detail in previous work in pyramidal cells (Narayanan and Johnston, J. Neurosci 2008) and the current study expands the work to OLM cells and in the context of in vivo-like synaptic inputs. Thus, the role of H-currents in regulating theta frequency firing is not entirely novel and the above study needs to be cited. Also, as detailed in Narayanan and Johnston, H-channel kinetics would be expected to contribute to resonance in the theta range.

2) In Figure 2, the more appropriate test for the -H condition would be to replace H with a leak conductance rather than adjusting the synaptic conductance levels. Especially since this was done on a per model basis and only to excitatory synapses, each model would receive a very different input. The authors need to consider replacing H with leak conductances and/or by bias currents rather than modifying synaptic conductances. How -H condition impacts membrane time constants also needs to be considered as this can impact intrinsic resonance of the model neuron. The same issue concerns the analysis of Kdr and KA effects as well, as again the membrane conductance is altered by channel closure and instead of changing leak conductance the authors re-tune synaptic conductance.

3) Comparison of models with somatic versus dendritic H-channel distributions in Table 2 shows a systematically higher membrane resistance and capacitance in models with somato-dendritic H-channels. This would be expected to change membrane time constants and could contribute to differences in the frequency preference among the 2 model classes. Controls are needed to exclude this possibility and confirm that H channel distribution underlies the differences between the models.

[Editors' note: further revisions were requested prior to acceptance, as described below.]

Thank you for resubmitting your work entitled "Computational models of O-LM cells are recruited by low or high theta frequency inputs depending on h-channel distributions" for further consideration at *eLife*. Your revised article has been favorably evaluated by Eve Marder (Senior Editor), a Reviewing Editor and two reviewers.

The manuscript has been improved but there are some remaining issues that need to be addressed before acceptance, as outlined below:

Please include the response figure as a supplemental figure, as it is an important control.

*Reviewer #1:*

The authors has addressed essential revision 1 in the manuscript. They have provided convincing explanations and additional simulations to address essential revisions 2 and 3 in the response letter but choose not to include the control simulations in the manuscript. I would recommend including a statement in the text related to Figure 2 that "control simulations in which an "artificial leak conductance" was introduced to maintain baseline firing without manipulating synaptic conductance yielded similar results" and all further simulation s were performed by scaling the excitatory synaptic conductance. They could refer reader to the response letter or add the Figure Figure 9 as a Supplement to Figure 2. In its current state, the issue of conductance raised by both reviewers is not addressed in the main manuscript.

*Reviewer #2:*

The authors addressed my points. I do not have any additional concerns.

---

## [Author Response]

*Essential revisions:*

*1) The role of H-currents and (M-type) potassium channels has been explored in detail in previous work in pyramidal cells (Narayanan and Johnston, J. Neurosci 2008) and the current study expands the work to OLM cells and in the context of in vivo-like synaptic inputs. Thus, the role of H-currents in regulating theta frequency firing is not entirely novel and the above study needs to be cited. Also, as detailed in Narayanan and Johnston, H-channel kinetics would be expected to contribute to resonance in the theta range.*

We acknowledge the point regarding previously published work on H-currents and theta resonance, including not only the mentioned Narayanan & Johnston (2008) work but also Hu et al. (2009) and others. It was not our intention to claim any fundamental groundwork from this particular perspective but rather to move such perspectives into the realm of hippocampal inhibitory interneurons, as almost no work has been done to date on H-current, theta resonance, and *inhibitory* cells. This is particularly pressing for O-LM cells as there is some controversy regarding their role as possible theta resonators or even pacemakers due to their expression of H-current, as we discussed in our Introduction (Rotstein et al., 20015; Kispersky et al., 2012; etc.). We have added the suggested references to the Introduction when discussing the background to H-current in O-LM cells.

*2) In Figure 2, the more appropriate test for the -H condition would be to replace H with a leak conductance rather than adjusting the synaptic conductance levels. Especially since this was done on a per model basis and only to excitatory synapses, each model would receive a very different input. The authors need to consider replacing H with leak conductances and/or by bias currents rather than modifying synaptic conductances. How -H condition impacts membrane time constants also needs to be considered as this can impact intrinsic resonance of the model neuron. The same issue concerns the analysis of Kdr and KA effects as well, as again the membrane conductance is altered by channel closure and instead of changing leak conductance the authors re-tune synaptic conductance.*

There are three issues to address in this essential revision: the possibility of adding bias current, adding a leak current, and the concerns of changes to the membrane time constants due to blocking various channels. We recognize that our approach of controlling for differences in membrane conductance due to the blocking of various channels by adjusting the excitatory synaptic conductances seems counter-intuitive at first. However, in our discussions regarding this work we had decided that adjusting excitatory synapses was the preferred method since the alternatives, namely adding a leak conductance and/or bias current, are less preferred for the following reasons.

Bias current. Adding a bias current would ideally be done by inserting a virtual current clamp in the soma; however, due to space clamp issues, the current spread may not extend fully into the distal dendrites of the models. Thus, the clamp could bias the results for one category of models over another, given that we are investigating the effect of dendritic H-current on spiking output in the context of rhythmic synaptic input modulation, also in dendrites. An alternative is to insert several electrodes in multiple dendritic locations. However, this already starts to resemble the case of adjusting excitatory synaptic inputs spread across the dendritic tree, and since we already had excitatory synaptic inputs, the need to add excitatory drive was more simply relegated to the pre-existing synaptic conductances. Furthermore, the added current clamp would require significantly different amounts of current depending on the model, given the differences in intrinsic channel properties of each model (see Table 1 in paper). To get a sense of how much current would be needed without running the full high-conductance simulations, we performed additional simulations where we blocked H-channels and fitted a somatic bias current needed to keep the cells at -74mV, the original conditions under which the models were developed, based on previous experimental voltage clamp data in O-LM cells (Sekulić et al., 2014). See the below table for the resulting bias current values for the models with H blocked; the table is a modified form of Table 2 from the manuscript, with only the “*I*_bias_” column kept, which is the fitted bias current for the intact model with H-current present. Note that the bias current was not turned on during our high conductance synaptic input simulations; the “*I*_bias_” column simply reports the values used when the original model database was developed in Sekulić et al. (2014).

**Somatic H****Somatodendritic H****Rank**
**Cell*****I*_bias_ (pA)*****I*_bias_ with H block (pA)****Rank**
**Cell*****I*_bias_ (pA)*****I*_bias_ with H block (pA)**3261-6.791.562251-10.316.65561-0.9327.423561-11.315.66131-1.576.789131-10.316.66201-0.9327.4212301-11.215.86891-8.190.15615201-10.516.57231-0.8467.5020501-10.516.57551-6.831.5221731-10.516.57691-8.160.18922861-10.416.5262-15.20.95362-7.09.80312-12.34.54342-7.049.78392-15.20.936372-6.99.90432-16.4-0.258492-7.049.77452-12.74.13572-5.7611.0602-12.64.20922-6.5610.2672-12.74.17962-2.1314.7682-15.20.901092-2.1814.6

Note the variability in required injected current values with H block, ranging from -0.258 pA to +16.6 pA. Because the injected current in the subsequent simulations would be held constant regardless of membrane potential conditions, the results could be more biased than adjusting the synaptic conductances, as follows. With conductance-based synaptic input, the amount of current flowing through the synapses varies as a function of the divergence of the membrane potential from the synaptic reversal potential. With current injection, however, the amount of current injected is held constant, regardless of the activity of the membrane potential. This might affect the subthreshold dynamics by providing more depolarizing current near threshold in models with more injected current that would not be the case if only conductance-based excitation was present. In the case of the latter, there is less excitatory current closer to threshold due to the reduced excitatory synaptic drive. Considering both the space clamp issues as well as the drawbacks of current-based injection described above, we decided that adjusting synaptic conductances would introduce fewer confounds. It is also a more physiologically plausible way to control for differences in membrane conductance due to blockade of voltage-gated channels.

Leak conductance. The possibility of adding an additional leak conductance avoids the problem of injecting currents over conductances. In the case of H-current block (“-H”), a depolarizing leak conductance would be required since the loss of inward current from the H-channels hyperpolarizes the membrane. The most straightforward implementation would be to insert a voltage-independent channel that carries the same reversal potential as our H-current channel model but with a different per-model conductance amount. This would be equivalent to reducing our H channel model to an “H leak” channel. By introducing a non-physiological channel, however, the comparison in spiking output would then be between a model with H current and a different one with an “H leak” channel, rather than the same model with and without H-current. The key consideration is that we believe that a test for the effect of H-current on spiking output should be between a control case with H-current, and one without H-current that also minimizes any other changes to the model.

Regarding both suggested changes to how we compensate for membrane hyperpolarization upon H block, we emphasize that in our simulations, we carefully adjusted *only* the excitatory conductances because the statistics of the excitatory drive were not changed during the simulations. Indeed, the result of the high bombardment of excitatory synapses is one of a conductance-based excitatory drive spread evenly across the dendritic tree. This method brings the membrane potential to the near-threshold fluctuating regime or high-conductance state while simultaneously avoiding both the space clamp and current-versus-conductance issues of a bias current, as well as the need to add a physiologically implausible artificial leak conductance. As for the inhibitory synaptic rate, since it was modulated as function of the input frequency, we therefore made sure not to change the inhibitory peak conductances as part of the compensation as this would have undoubtedly biased the results of the resulting spiking activity. These considerations apply equally to the potassium currents in the model that were blocked.

The overriding issue is that, given the heterogeneity of the models used (see Table 1 in paper), *some* parameter will need to vary for each model to compensate for membrane hyperpolarization upon H block. Whether it is excitatory peak conductance, bias current, or artificial leak conductance, the modifications made to each model will not be identical. The key issue, then, is not whether some parameter will be very different across models (it necessarily must be), but rather whether the modified parameters unduly affect the variable of interest or not(viz., spiking resonance as measured by power ratio, rotation number, etc.). Adding current injection may affect the spiking resonance as discussed above and is therefore not a good way to control for channel block

The question then arises, how do we ensure that the resulting changes to spiking resonance are comparable, in other words, that we are comparing “apples to apples” with the control case and the “treatment” cases of H block, Kdrf block, or KA block? We took the precedent of Kispersky et al. (2012) where all models are situated in the baseline condition of being in the high-conductance state, as defined by an average of 2.5 Hz firing without input modulation, while maximizing the subthreshold fluctuations. It is not possible to fully control for subthreshold dynamics as these will differ depending on the model’s intrinsic channel complement, which is to be expected. Rather, we ensure the baseline spiking activity does not vary across models. Therefore, the baseline measure for ensuring a proper comparison between control and treatment cases is the spike rate with no modulation. This can be seen in Figure 2—figure supplement 1 (as well as the new Figure 4—figure supplement 1), where the firing rate for all models hovers around 2.5Hz in the control (*H*_S_ and *H*_SD_) and various H block (-H; -H -Kdrs; -H -KA) conditions. Note that any variability in spike rates between the conditions is strictly due to the granularity of the excitatory peak conductance parameter search space because we chose the parameter value such that 2.5 Hz firing would be achieved. This was a tradeoff between finding as close to 2.5 Hz firing as possible versus the added computational costs of repeated sweeps of finer and finer spaced values of the peak excitatory conductance for each model. Accordingly, any changes to spiking upon introduction of a modulation of the inhibitory synaptic input can be explained by an appeal to the intrinsic channel complement, whether in control or H block, etc.

Despite the considerations regarding the artificial leak conductance described above, we did additional sets of input modulation simulations on high-performance computers (as per the methods described in our paper) in order to verify that our key results would not change if we had kept the excitatory synaptic peak conductances fixed and instead added a leak channel. We implemented the “H leak” channel as described above. We fitted the conductance density by taking as a baseline each model’s H maximum conductance parameter, then scaling it by the value of the H steady state activation function at the mean of the subthreshold *V*_m_ fluctuations in the control case with H intact. This still provided more depolarizing current for most models, and a parameter search for reducing the H leak conductance was performed. We then ran all synaptic input modulation simulations for three cases: H leak only; H leak with -Kdrs; H leak with -KA. The H leak conductance had to be refitted for each of the cases to arrive at the same baseline 2.5 Hz firing with no modulation of the inputs, resulting in the high-conductance state. Crucially, we did not change the excitatory synaptic conductance from the control case. As in our present paper, we found that the “H leak” models resulted in the same partitioning of model responses into low and high theta dependent on dendritic distribution (Figure 2—figure supplement 2).

For the case of H leak with Kdrs block, we found that the models originally with somatodendritic H, now replaced by “H leak”, could not have their firing rates sufficiently reduced to the 2.5Hz baseline firing with no modulation by adjusting the “H leak” conductance density alone – even when setting it to zero. Thus, further reduction of the firing rate without changing the excitatory synaptic peak conductances would have required adjusting the maximum conductance density of another channel entirely (e.g., lowering calcium channel conductances or increasing the M-channel potassium conductance). Because this would shift the balance of model parameters in a way that would have potentially introduced further confounds, we did not do additional -Kdrs simulations with unchanged synaptic conductances. However, we believe that the similarity of our results with replacing H blocked (-H) models with “H leak” models as shown in Figure 2—figure supplement 2, sufficiently demonstrates the validity of our initial approach of adjusting the peak excitatory conductance only to compensate for changes in baseline firing characteristics of the models under the various channel blocking conditions.

Intrinsic resonance and channel block. The reviewers further express the concern that blocking H, as well as Kdrs and KA, will affect the membrane time constant and thus affect the intrinsic resonance of the models. However, intrinsic resonance is a function not only of the membrane time constant (i.e., passive properties of a cell) but also active channels (Hutcheon and Yarom, 2000). Indeed, the passive membrane as represented by an RC circuit (with a leak and capacitor in parallel) only attenuates responses to high frequencies, i.e., acts as a low-pass filter (Koch 1999). On the other hand, attenuation at lower frequencies depends on currents that oppose changes in membrane voltage (Hutcheon and Yarom, 2000). This can only be performed by voltage-gated channels that are selective to ions with electrochemical gradients such that upon opening, a current is produced that opposes the *V*_m_ trajectory. The frequency selectivity of the resulting resonance depends on the kinetics of the particular active channel(s) under consideration. Thus, the change in intrinsic resonance upon channel block essentially underlies what is being examined in our simulations. However, we did not try to thoroughly investigate the subthreshold resonance directly (e.g., using Q value as per Hu et al., 2009) because subthreshold resonance does not necessarily translate to supra-threshold or spiking resonance, although it is not necessarily an independent phenomenon either. For instance, Kispersky et al. (2012) found that with an added artificial *I*_h_ current, despite their O-LM cells exhibiting strong subthreshold resonance, this did not translate into *I*_h_-dependent spiking resonance. In our paper, we focus on the spiking resonance as a function of input modulation and channel complement. Spiking O-LM cells (at theta frequencies) would influence pyramidal cells and thus play a role in theta rhythms. The spiking responses of our models exhibit a band-pass characteristic where in general low (< 2Hz) and high (>16 Hz) frequency-modulated inputs do not entrain spiking, but only high and low theta, depending on H distribution.

We decided not to include the additional simulations and above detailed response in our revised manuscript. Rather, we have added a few sentences to alert the reader to these considerations. Also, since *eLife* publishes these responses, interested readers would have direct access to these details.

*3) Comparison of models with somatic versus dendritic H-channel distributions in Table 2 shows a systematically higher membrane resistance and capacitance in models with somato-dendritic H-channels. This would be expected to change membrane time constants and could contribute to differences in the frequency preference among the 2 model classes. Controls are needed to exclude this possibility and confirm that H channel distribution underlies the differences between the models.*

The considerations in our response to essential revision 2 regarding how the membrane time constant relates to intrinsic resonance largely applies here as well – namely, that changes in *τ*_m_ may not affect resonance appreciably, especially with regard to the differential attenuation of lower frequencies we see in our models, resulting in high (4-9 Hz) or low (2-5 Hz) theta responses. However, the differences in *R*_m_ are noted and should be accounted for – we appreciate the reviewers noticing this so that we can address it, and thus strengthen our work.

In considering this, we note that the large differences in *R*_m_ only pertain to the models with morphology 1 (“Cell 1”, upper 8 rows in Table 2). The models with morphology 2, on the other hand, do not exhibit very large differences in *R*_m_ after passive property re-fitting. Given this, we can separate our analyses for morphology 1 and 2 and determine whether our results still hold. That is, essentially using morphology 2 as a control for testing the effects of changing *R*_m_ on spiking resonance.

The reason for the difference can be addressed by noting that the surface area for the two morphologies is 16,193.6 µm^2^ for cell 1 and 9,980.1 µm^2^ for cell 2 (Sekulić et al., 2015). Nevertheless, for each class of model selected from the database, a constant maximum conductance density for H was chosen (0.5 for somatic H and 0.1 for somatodendritic H). This results in overall higher levels of total membrane conductance due to H for models with cell 1 compared to cell 2. Since the experimental recordings we used in fitting the passive properties at the time were not performed with blockade of active channels, we needed to keep the channels intact in our models as well. Because H is active in the clamped hyperpolarized regime, the models with the morphology 1 were leakier when H was in the dendrites, compared to models with morphology 2, due to more total H conductance. This meant that to keep the membrane time constant fixed to match the recordings, the intrinsic leak conductance needed to be lowered in compensation. Since *R*_m_ is the inverse of the leak, a smaller leak would then result in a higher R_m_ value for models with somatodendritic H and morphology 1. The small increase in *R*_m_ with models with morphology 2 with somatodendritic H was also due to a small increase in total H conductance in the somatodendritic H compared to somatic H models. The below table showing total H conductance for each of the four classes of models (per morphology, per H distribution) is consistent with this explanation:

**Total membrane H conductance (nS)*****H*_S_*****H*_SD_****Cell 1**0.1420.281**Cell 2**0.1630.166

To show that the larger increase in *R*_m_ with models with morphology 1 do not affect the results of the paper, we separated out the models with each morphology and re-analyzed their power ratios. The results can be seen in Figure 9.

Author response image 1.Partitioning of spiking responses of O-LM models into high and low theta when separately analyzing models with different morphologies.Models with morphology 1 (**A**) or morphology 2 (**B**) and HS (left) vs HSD (right) distributions. Statistical test used was repeated measures ANOVA for the populations of *H*_S_ and *H*_SD_ models between all modulation frequencies crossed with H condition (morphology 1 *H*_S_: *F*_(1,15)_ = 2.23, p=0.01, n=8; morphology 1 *H*_SD_: *F*_(1,15)_ = 0.75, p=0.64, n=5; morphology 2 *H*_S_: *F*_(1,15)_ = 2.31, p=0.37, n=8; morphology 2 *H*_SD_: *F*_(1,15)_ = 2.60, p=0.02, n=7; Huynd-Feldt correction reported for all tests). Boxplot annotations as per Figure 1 legend. Note that not enough models with morphology 2 and somatic H completed the modulation simulations under 48 hours for frequencies above 12 Hz (B, left) to be able to perform a repeated measures ANOVA test; thus, only simulations with modulation frequencies 12 Hz and below are included in the analysis.**DOI:**
http://dx.doi.org/10.7554/eLife.22962.020

Then, using the same criteria as in the manuscript for delineating the peak spiking response (viz., power ratio > 0.6), we find that models with cell 1 exhibit a partitioning of responses between somatic H (2-8Hz, with a peak at 3Hz, control) and somatodendritic H (4-20Hz, with a peak at 8Hz, control). Models with cell 2 also exhibit a partitioning of responses between somatic H (2-4Hz, with a peak at 3Hz, control), and somatodendritic H (4-12Hz, with a peak at 4Hz, control). Although the upper end of each frequency range differs between the morphologies, the overall partitioning into lower and higher theta responses can still be seen, given that theta is often reported as being anywhere within the 4-12Hz range. The lower cut-off of the models with somatic H not responding to lower than 2Hz inputs, and somatodendritic H models not responding to lower than 4Hz inputs, is the same for models of either morphology. Thus, we conclude that the apparent differences in *R*_m_ between models with the two morphologies do not affect the main results presented in the paper.

We decided not to incorporate these details in the revised manuscript but, as with Essential revision 2, the interested reader will be able to access this response directly.

[Editors' note: further revisions were requested prior to acceptance, as described below.]

*The manuscript has been improved but there are some remaining issues that need to be addressed before acceptance, as outlined below:*

*Please include the response figure as a supplemental figure, as it is an important control.*

Figure 2—figure supplement 2 appears in the revised manuscript, and the figure has been uploaded separately as well.